# New [10]Be exposure ages improve Holocene ice sheet thinning history near the grounding line of Pope Glacier, Antarctica

Jonathan R. Adams[1,2], Joanne S. Johnson[1], Stephen J. Roberts[1], Philippa J. Mason[2], Keir A. Nichols[2], Ryan A. Venturelli[3], Klaus Wilcken[4], Greg Balco[5], Brent Goehring[6], Brenda Hall[7], John Woodward[8], Dylan H. Rood[2]

1 British Antarctic Survey, High Cross, Madingley Road, Cambridge CB3 0ET, UK

2 Department of Earth Science & Engineering, Imperial College London, London SW7 2AZ, UK

3 Department of Geology and Geological Engineering, Colorado School of Mines, Golden CO 80401, USA

4 Australia's Nuclear Science and Technology Organisation (ANSTO), New Illawarra Road, Lucas Heights, NSW 2234, Locked Bag 2001, Kirrawee DC 2232, Australia

5 Berkeley Geochronology Center, 2455 Ridge Road, Berkeley, CA 94709, USA

6 Department of Earth & Environmental Sciences, Tulane University, New Orleans, LA 70118, USA

7 School of Earth and Climate Sciences and the Climate Change Institute, University of Maine, Orono, ME 04469 USA

8 Department of Geography and Environmental Sciences, Northumbria University, Newcastle-upon-Tyne, NE1 8ST, UK

*Correspondence to*: Jonathan R. Adams (j.adams19@imperial.ac.uk)

Co-author email list (alphabetical order):
Greg Balco – balcs@bgc.org
Brent Goehring – bgoehrin@tulane.edu
Brenda Hall – brendah@maine.edu
Joanne S. Johnson – jsj@bas.ac.uk
Philippa J. Mason – p.j.mason@imperial.ac.uk
Keir A. Nichols – keir.nichols@imperial.ac.uk
Stephen J. Roberts – sjro@bas.ac.uk
Dylan H. Rood – d.rood@imperial.ac.uk
Ryan A. Venturelli – venturelli@mines.edu
Klaus Wilcken – klausw@ansto.gov.au
John Woodward – john.woodward@northumbria.ac.uk

**Abstract.** Evidence for the timing and pace of past grounding line retreat of the Thwaites Glacier system in the Amundsen Sea embayment (ASE) of Antarctica provides constraints for models that are used to predict the future trajectory of the West Antarctic Ice Sheet (WAIS). Existing cosmogenic nuclide surface exposure ages suggest that Pope Glacier, a former tributary of Thwaites Glacier, experienced rapid thinning in the early- to mid-Holocene. There are relatively few exposure ages from the lower ice-free sections of Mount Murphy (< 300 m asl) that are uncomplicated by either nuclide inheritance or scattering due to localised topographic complexities; this makes the trajectory for the latter stages of deglaciation uncertain. This paper presents 12 new $^{10}$Be exposure ages from erratic cobbles collected from the western flank of Mt Murphy, within 160 m of the modern ice surface and 1 km from the present grounding line. The ages comprise two tightly clustered populations with mean deglaciation ages of 7.1 ± 0.1 ka and 6.4 ± 0.1 ka (1SE). Linear regression analysis applied to the age-elevation array of all available exposure ages from Mt Murphy indicates that the median rate of thinning of Pope Glacier was 0.27 m yr$^{-1}$ between 8.1–6.3 ka, occurring 1.5 times faster than previously thought. Furthermore, this analysis better constrains the uncertainty (95 % confidence interval) in the timing of deglaciation at the base of the Mt Murphy vertical profile (~80 m above the modern ice surface), shifting it to earlier in the Holocene (from 5.2 ± 0.7 ka to 6.3 ± 0.4 ka). Taken together, the results presented here suggest that early- to mid-Holocene thinning of Pope Glacier occurred over a shorter interval than previously assumed and permit a longer duration over which subsequent late Holocene rethickening could have occurred.

## 1 Introduction

The Amundsen Sea Embayment (ASE), dominated by the Pine Island – Thwaites Glacier system, has recently undergone the fastest rates of ice mass loss of all sectors of the West Antarctic Ice Sheet (WAIS), which is estimated at 136 Gt yr$^{-1}$ (70 % from Pine Island – Thwaites) from 2009–2017 (Rignot et al., 2019). Ice mass loss from WAIS is driven principally by high rates of basal melting caused by incursions of warm Circumpolar Deep Water (CDW) onto the continental shelf (Adusumilli et al., 2020; Pritchard et al., 2012). This basal melting has led to increased ice flow velocity (Rignot et al., 2014), faster grounding line retreat (Konrad et al., 2018; Milillo et al., 2022) and dynamic ice thinning (Pritchard et al., 2009; Shepherd et al., 2019). However, quantifying how much WAIS will contribute to future global mean sea level rise under different emissions pathways remains subject to considerable uncertainty (Bamber et al., 2019, Oppenheimer et al., 2022). Physics based ice sheet models simulating evolution of WAIS (e.g. (Albrecht et al., 2020; Pollard and DeConto, 2009)) require geologic records over centennial – millennial timescales for validation (Bentley, 2010). However, existing geologic records of ice sheet change since the Last Glacial Maximum (LGM) in the ASE based on cosmogenic nuclide surface exposure ages (Johnson et al., 2008, 2014, 2017, 2020; Lindow et al., 2014) are either incomplete or exhibit considerable scatter closest to the modern ice surface. Here we present new $^{10}$Be surface exposure ages from erratic cobbles that improve the thinning history of the lowest 300 metres of presently exposed rock at Mount Murphy, a volcanic edifice adjacent to Pope Glacier in the central ASE (Fig. 1a). These data fill a critical gap in the Holocene ice sheet thinning history of Pope Glacier.

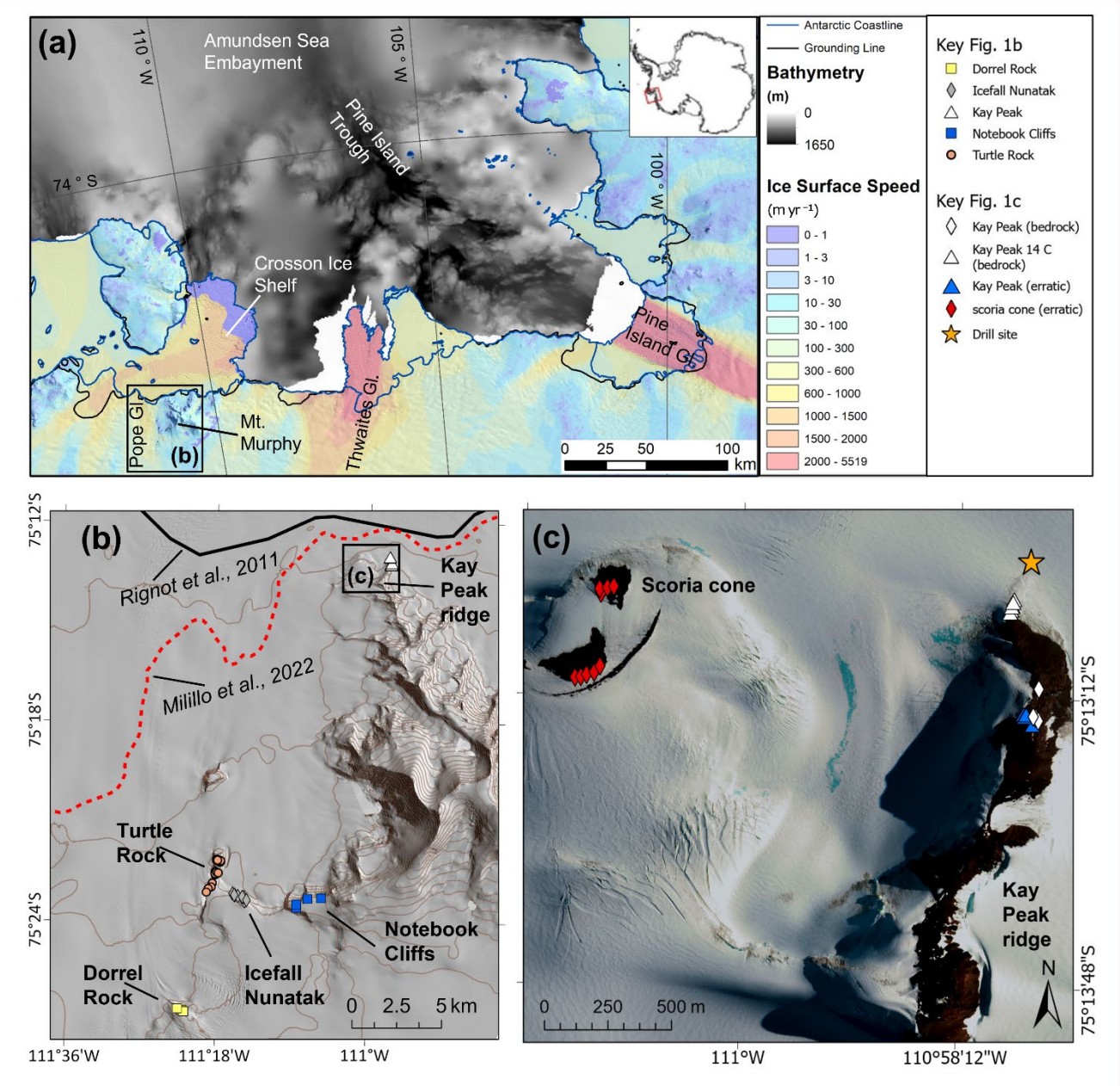


**Figure 1: Location of the study area in the context of Mt Murphy and the wider ASE.** (a) ASE area map displaying regional ice streams, ice velocities (Mouginot et al., 2012) overlayed on Reference Elevation Model of Antarctica (REMA) (Howat et al., 2019) regional bathymetry (Arndt et al., 2013), 2011 grounding line (black line; Rignot et al., 2011) and Antarctic coastline (blue; Antarctic Digital database, version 7.2). (b) Mt Murphy site map generated using REMA derived hill shade, with contours representing 100 m intervals. Locations of
previous exposure ages (Johnson et al., 2020, 2008) indicated by coloured symbols shown in Fig. 1b key, also displayed are both the 2011 grounding line (same as in Fig. 1a), and most recent localised grounding line (red dotted; Milillo et al., 2022). (c) High resolution satellite imagery (732 false colour composite), DigitalGlobe Products, WorldView 2 © 2013 DigitalGlobe, Inc., a Maxar company, showing scoria cone and Kay Peak Ridge. Drill site indicates subglacial bedrock core recovery by the Geological History Constraints (GHC) subglacial drilling project (Boeckmann et al., 2021).

This study applies the method of cosmogenic nuclide surface exposure dating to determine the timing and rate of ice surface elevation change since the Last Glacial Maximum (LGM). This ice surface history is achieved by measuring the cosmogenic nuclide content of bedrock and glacial deposits, such as cobbles or boulders, which have undergone englacial transport (Ackert et al., 1999; Stone et al., 2003; Johnson et al., 2014). Several cosmogenic surface exposure studies reconstructing post LGM changes in ice surface thickness have been conducted in the ASE (Lindow et al., 2014; Johnson et al., 2008, 2014, 2020).

These previous studies have improved our understanding of the post-LGM deglaciation history of outlet glaciers in the central and eastern embayment. From these previous studies, we have a relatively good understanding of the trajectory of ice sheet thinning after the LGM until the mid-Holocene, but few exposure age arrays extend close to the modern ice sheet surface. At Mt Murphy, where a high density of samples from higher elevations have been analysed, exposure ages at elevations within < 200 m of the modern ice surface are either absent or scattered (fig. 5, Johnson et al., 2020). This age uncertainty makes it

difficult to determine whether the rapid thinning indicated by the higher elevation exposure age data slowed down before reaching its modern elevation, or whether the fast rate of thinning observed between 9–6 ka continued after that time. Furthermore, the time when the ice surface reached its modern elevation cannot be determined from the existing data because no records of thinning in the last 5 ka in the ASE currently exist from above the modern ice surface (Johnson et al., 2014, 2020; Lindow et al., 2014). This existing data is ambiguous and implies either that ice thickness - and by inference grounding

line position of Pope Glacier - has been largely stable over this period, or that any evidence for late Holocene thinning/retreat is currently below the present ice surface (Johnson et al., 2022).

To improve our understanding of Holocene ice sheet history in the Pine Island – Thwaites Glacier region, we focus here on the area within 1 km of the grounding line at Mt Murphy (Milillo et al., 2022) (Fig. 1). The base of Kay Peak ridge was the

location for a subglacial bedrock drilling campaign undertaken in 2019–20 by the International Thwaites Glacier Collaboration (Fig. 1c). The aim of that campaign was to collect bedrock cores and measure cosmogenic nuclide concentrations within them to detect whether the ice sheet surface was ever lower than present in the past few millennia. Here we present new $^{10}$Be surface exposure ages for 12 erratic cobbles collected from the surface of a scoria cone situated 1.6 km west of the drill site. The current understanding of the thinning history of Pope Glacier and mid- to late-Holocene ice sheet configuration in the ASE is

improved by dating erratic cobbles from this site for the following three reasons. First, the lower section of the Mt Murphy vertical thinning transect is currently poorly constrained, largely due to an absence of exposure ages from 240–100 m above the modern ice surface, equivalent to 320–180 metres above sea level (m asl). There is also spread in the array of existing exposure ages (both $^{10}$Be and to a lesser extent $^{14}$C) from Kay Peak at the base of the profile 100–80 m above the modern ice surface (see fig. 5, Johnson et al., 2020). The scoria cone is situated at the ideal elevation (180–240 m asl) to fill this data gap

and better constrain the deglacial history. Second, the scoria cone site is situated in close proximity (0.7 km) to the current grounding zone of Pope Glacier (Fig. 2) (Milillo et al., 2022). Thus, this site is expected to be highly sensitive to present and past changes in grounding line position during the mid- to late-Holocene. Finally, exposure ages from the site will directly inform interpretation of the cosmogenic nuclide concentrations within the subglacial bedrock cores that were recovered from

the projection of Kay Peak ridge under Pope Glacier, which are critical to determining if the ice surface was thinner than present at any time in the last 5 ka.

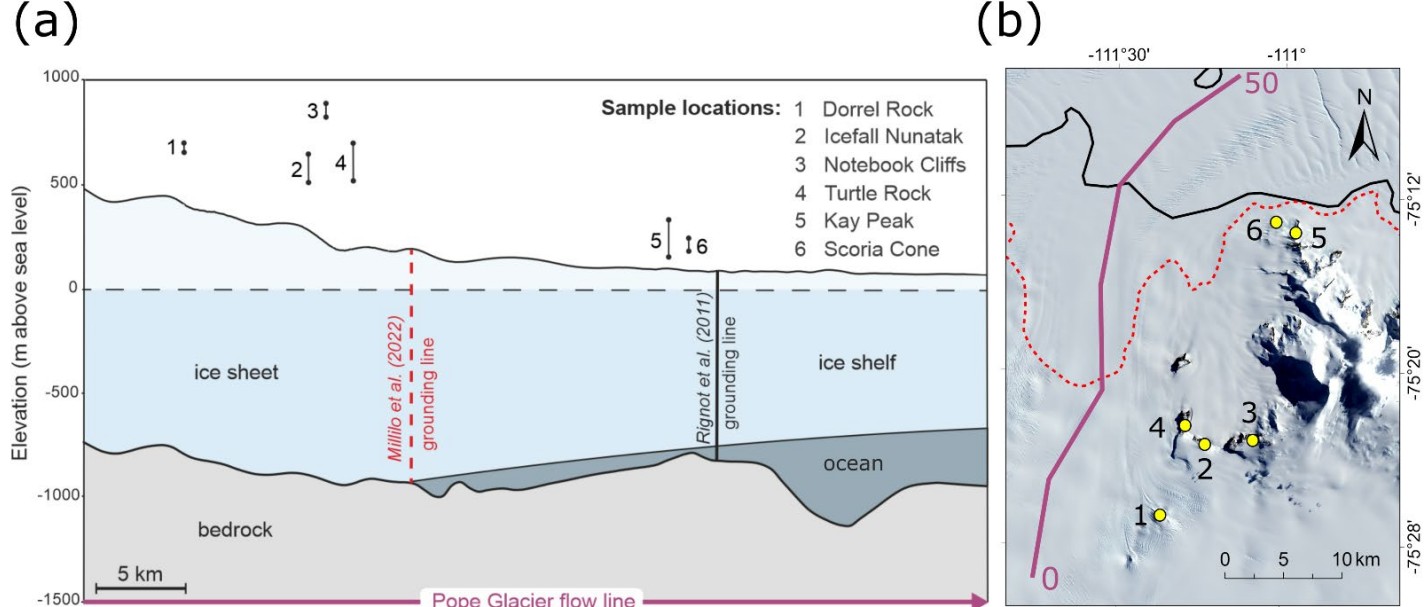

**Figure 2: Relationship between nunatak elevation, bedrock and ice sheet topography** (a) Cross-section transects showing modern ice surface elevation data (line above sea level; from Reference Elevation Model for Antarctica (REMA); (Howat et al., 2019) and bedrock topography (line below sea level; from BedMachine; (Morlighem et al., 2020) . Note that the glacier flowline profile (solid purple line in panel b) is >10km from the sample sites; elevation range markers on the cross-section represent the highest and lowest elevation sample at each site (m asl) (Johnson et al., 2020) and are displayed at points approximately adjacent to flowline on the right hand panel. (b) Landsat-8 true colour composite (432) image showing location of Pope Glacier flowline with distance downstream from arbitrary starting position. Profile is displayed in the cross section above. Elevation range markers on the cross-section (a) are displayed as yellow dots approximately adjacent to flowline on panel b. Landsat-8 image courtesy of the U.S. Geological Survey; https://doi.org/10.5066/P9OGBGM6.

## 2. Site Description and Methods

### 2.1 Site Description

Mt Murphy, which reaches a maximum elevation of 2703 m asl, is located approximately 50 km from the modern Thwaites Glacier ice stream and is bounded by Crosson Ice shelf to the north and Pope Glacier to the west (Johnson et al., 2020) (Fig. 1). Pope Glacier is approximately 14 km wide and flows into the Crosson Ice Shelf at a velocity of 0.8 km yr$^{-1}$ (Mouginot et al., 2012). Mt Murphy hosts an abundance of glacial deposits including numerous erratics and striated bedrock surfaces. These erratics and striations indicate that glacial ice previously covered and flowed over the lower section of Mt Murphy including scoria cone and Kay Peak ridge (Johnson et al., 2020).

We collected 12 glacially deposited erratic cobbles for cosmogenic surface exposure age dating from two outcrops on the
scoria cone, which is situated < 1 km from the present grounding line of Pope Glacier (Milillo et al., 2022) (see Fig. 1b). The
two bedrock outcrops onto which the erratics were deposited, hereafter referred to as outcrop A (upper) and outcrop B (lower),
mostly consist of rubbly oxidised scoria accompanied by smaller exposures of hyaloclastite breccia. The outcrops form a
basaltic landform of unknown age, that is a parasitic cone on the main Mt Murphy volcanic shield.

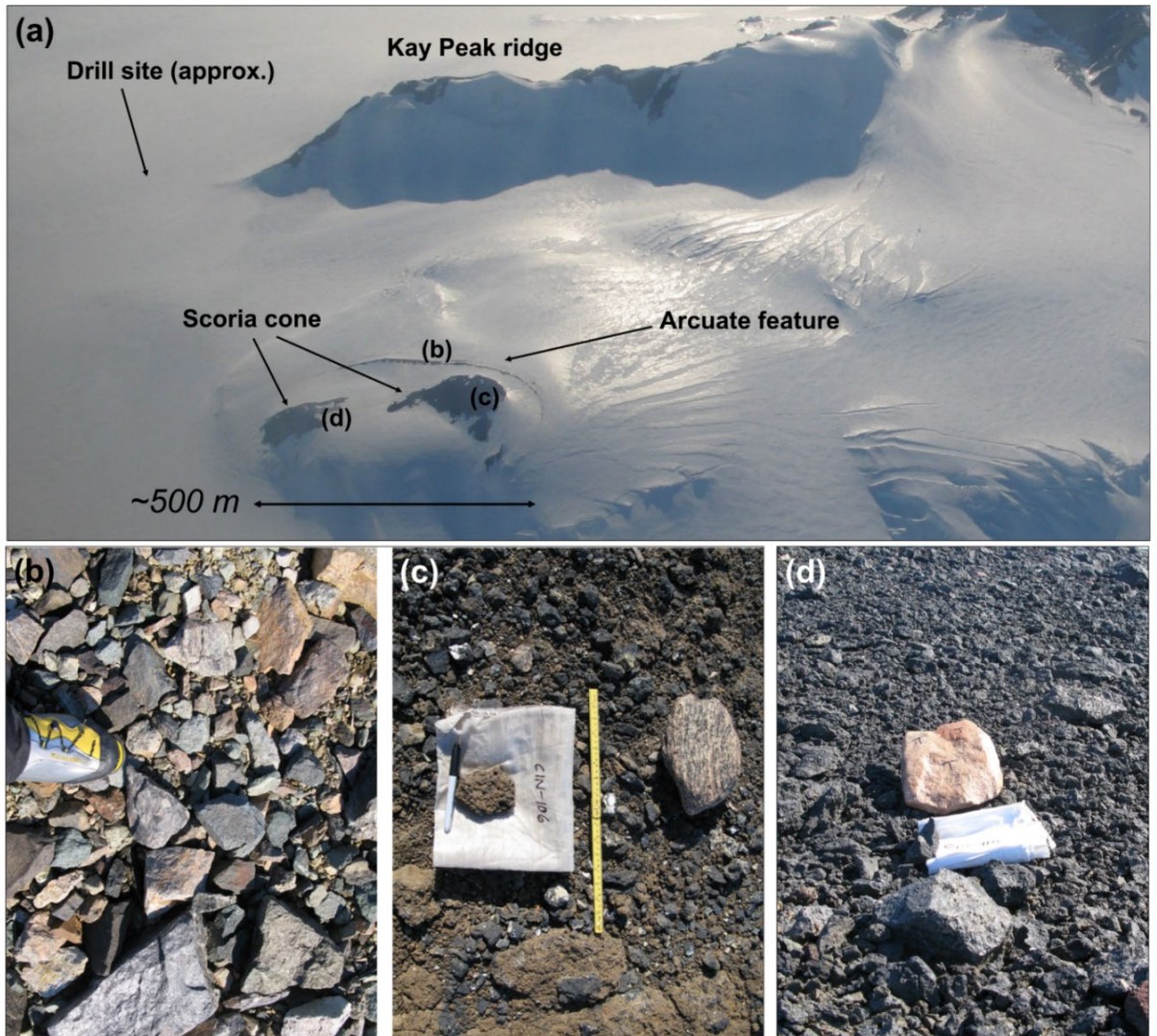

**Figure 3: Geomorphic difference between clasts deposited at the scoria cone.** (a) Image showing location of scoria cone site in relation
to Kay Peak ridge. Horizontal black line indicates approximate field of view. (b) Clasts forming the arcuate ridge landform. (c and d) Erratic
cobbles (samples CIN-106 and CIN-110) perched on the bedrock of scoria cone. Approximate locations of clasts displayed in (b – d) shown
in top panel. Scale indicated by boot (b), 50 cm ruler (c) and sample bag (d). Images of all scoria cone erratics collected for exposure dating
are provided in the supplementary material (Fig. S3). Photo credit: Joanne S. Johnson and Stephen J. Roberts.

The samples collected from the scoria cone range in altitude from 178–239 m asl, which equates to an elevation of ~100–160 m above the modern ice surface. The position of each sample was recorded using a Trimble 5700 GPS receiver set at the same height as the sample upper surface. Height above the ellipsoid was corrected to orthometric height (height above geoid EGM08) using Precise Point Positioning in Bernese software (see Johnson et al., 2020). The modern ice surface elevation used in the present paper was extracted from a digital elevation model (DEM) of Mt Murphy (see Johnson et al., 2020, Supplementary

Material). Topographic profiles illustrating the elevation and position of the scoria cone outcrops and samples relative to the modern ice surface can be found in Appendix C.

Erratic cobbles, primarily composed of gneiss and granite rock types, are present on the surface of both outcrops and provide evidence that the site was previously overridden by ice. Of the 12 erratic cobbles that we analysed, 6 are medium cobbles (long

axis 15–50 cm) and 6 are small cobbles (long axis < 15 cm). Half the cobbles (n = 6) were sub-rounded and only one of the cobbles collected was angular (CIN-111) (Table S2). The characteristics of erratics deposited on scoria cone, such as clast shape and lithology, differ from those of material deposited at a nearby landform that forms an arcuate ridge (Fig. 1c, Fig. 3). Material from this landform is predominantly angular, indicating it is locally derived (probably from Kay Peak), whilst erratics deposited at scoria cone are more rounded. The roundness of many erratics emplaced at scoria cone is evidence that they were

subject to prolonged erosion during englacial transport (Darvill et al., 2015). Moreover, six of the erratics at scoria cone are granitic (granite/aplite), a rock type not common on Mt Murphy or surrounding nunataks. Erratics of similar shape and rock type have been observed at other nunataks surrounding Mt Murphy [Turtle Rock and Icefall Nunatak (Fig. 1; Johnson et al., 2020)], indicating that the glacial deposits selected for [10]Be analysis from scoria cone originated from a source upstream rather than being locally derived from Kay Peak ridge.


## 2.2 Analytical Methods

Quartz mineral separation and [10]Be isotope dilution chemistry was conducted in the CosmIC Laboratory at Imperial College London using standard procedures (Kohl and Nishiizumi, 1992; Corbett et al., 2016). After beryllium was extracted and purified from quartz, the samples were loaded into cathodes for [10]Be measurement by accelerator mass spectrometry (AMS).

These AMS measurements were performed at the Australian Nuclear Science and Technology Organisation (ANSTO) (Wilcken et al., 2017). Measurements were then normalised to the KN-5-3 standard with an assumed ratio of 6.320 x $10^{-12}$ ($t_{1/2}$ = 1.36 Ma; (Nishiizumi et al., 2007)). Exposure ages were calculated using version 3 of the online calculators at hess.ess.washington.edu (Balco et al., 2008). The online calculators use the LSDn production rate scaling method for neutrons, protons and muons following Lifton et al., 2014 and summarised in Balco, 2017 and the primary production rate calibration

data set of (Borchers et al., 2016). In order to keep the input parameters consistent with those used by Johnson et al., 2020, all exposure ages are reported assuming no erosion or snow cover and a material density of 2.7g $cm^{-3}$.

## 3. Results

The [10]Be exposure ages obtained from the scoria cone range from 7.5 ± 0.5 ka to 6.2 ± 0.4 ka (1σ external errors on individual ages throughout, unless otherwise noted, e.g., Fig. 4), and the average exposure age is 6.8 ka (n=12) (Table 1). The exposure ages are clustered in two separate groups (Fig. 4a), which correspond to outcrops A and B (Fig. 4b). Outcrop A contains a cluster of ages at an elevation range 225–239 m asl, whereas outcrop B ages are all clustered at a narrower elevation range of ~180 m asl. We interpret the ages as representing the timing of the most recent deglaciation of the two scoria cone outcrops.

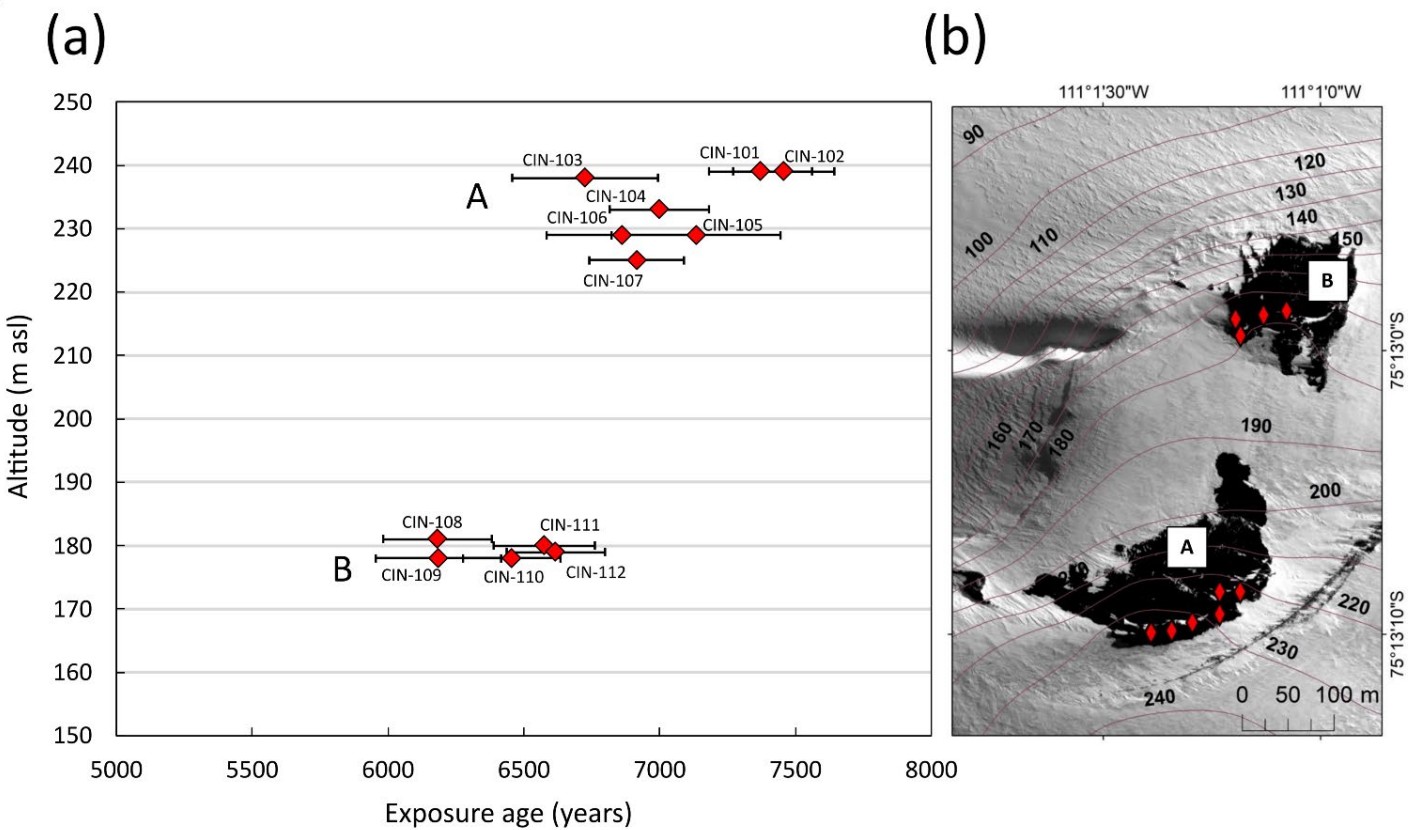

**Figure 4: Exposure ages from the two scoria cone outcrops.** (a) [10]Be exposure ages versus sample altitude. Error bars displayed represent internal uncertainty of AMS measurement on samples (1σ). (b) Very high-resolution panchromatic satellite image of scoria cone site, credit; DigitalGlobe Products, WorldView 2 © 2013 DigitalGlobe, Inc., a Maxar company. Purple elevation contour lines in units of metres above sea level. Red diamonds represent the location of each sample. The separate outcrops are labelled as A and B and correspond to the upper and lower population clusters displayed in panel a, respectively. Note only 10 sample locations are displayed on panel b as some of the samples were so close to each other they were recorded with the same GPS location. For topographic profiles illustrating the scoria cone outcrops relative to the modern ice surface, see Fig. C1 (Appendix). See Table S1 for analytical data.

The two clusters of ages each correspond to a distinct outcrop; the higher elevation cluster (n = 7) were all obtained from outcrop A and the lower elevation cluster (n = 5) from outcrop B (Fig. 4b). The reduced $\chi^2$ ($\chi^2v$) and probability (p) values for ages from each outcrop (outcrop A: $\chi^2v$ = 1.67, p value $\geq$ 0.01; outcrop B: $\chi^2v$ = 1.11, p value $\geq$ 0.01) indicate that the scatter in exposure ages at each outcrop can be attributed to analytical uncertainty alone (Balco, 2011). However, when the same statistical test is performed across both outcrops the reduced $\chi^2v$ value is 4.28 and p value is < 0.01. Taken together, these statistical analyses are consistent with the interpretation that the ages from outcrop A and B are two statistically different populations and represent distinct times of deglaciation. Therefore, the error weighted mean standard error provides the best estimate for the outcrop deglaciation ages: 7.1 $\pm$ 0.1 ka for outcrop A and 6.4 $\pm$ 0.1 ka for outcrop B.

| Sample Group Name | No. of samples (n) | Sample range elevation (m asl) | Sample range elevation above modern ice (m) | Error weighted mean (ka) | Std. error (yrs.) | Ext. error (yrs.) | Reduced $\chi^2$ ($\chi^2v$) | p value |
|---|---|---|---|---|---|---|---|---|
| Scoria cone outcrop A | 7 | 239 - 225 | 159 - 145 | 7.1 | 80 | 426 | 1.67 | 0.1240 |
| Scoria cone outcrop B | 5 | 181 - 178 | 101 - 98 | 6.4 | 86 | 390 | 1.11 | 0.3521 |
| Scoria cone (comb.) | 12 | 239 - 178 | 159 - 98 | 6.8* | | | 4.28 | 0.0000 |
| Lower Kay Peak ([14]C) | 6 | 170 - 150 | 100 - 80 | 5.8* | | | 15.37 | 0.0000 |
| Rev. lower Kay Peak ([14]C) | 4 | 167 - 154 | 97 - 84 | 6.5 | 252 | 303 | 3.71 | 0.0110 |

**Table 1: Summary statistics of new [10]Be exposure ages from scoria cone and previously published [14]C Kay Peak ridge bedrock ages.** Table includes error weighted mean exposure age, reduced $\chi^2$ ($\chi^2v$) value, and p value from each sample group. When p value > 0.01, the error weighted mean value and standard error is reported for that statistically significant population. When the reduced $\chi^2v$ p value is < 0.01, the arithmetic mean value indicated by * is provided without an associated uncertainty because the reduced $\chi^2v$ and p value are not consistent with a single statistically significant population (see Results section for details). The sample elevation range above the modern ice surface is relative to the ice sheet elevation adjacent to Kay Peak ridge reported in Johnson et al. (2020) and using 80 m asl as the modern ice surface elevation at scoria cone site.

We now compare the new scoria cone exposure ages to previously published in situ [14]C bedrock exposure ages from lower Kay Peak ridge (Fig. 1c). The existing six Kay Peak ridge samples measured for in situ [14]C are from a similar elevation to the scoria cone samples. They range from 100–80 m above the modern ice surface (170–150 m asl) (Johnson et al., 2020), with exposure ages spread between 8.0 $\pm$ 0.6 ka and 3.7 $\pm$ 0.3 ka. The range in elevation of samples measured for [10]Be from scoria cone is similar, 160–100 m above the modern ice surface (240–180 m asl), but the age range is smaller than at lower Kay Peak ridge (7.5 $\pm$ 0.4 ka to 6.2 $\pm$ 0.5 ka, n=12). The average in situ [14]C age for the lower Kay Peak ridge samples is 5.8 $\pm$ 1.4 ka (1SD, n=6; Johnson et al., 2020), and they have $\chi^2v$ = 15.37 and p value < 0.01 (Table 1). This indicates that the six in situ [14]C ages have scatter above that expected from analytical uncertainties alone and, therefore, are not from a single statistically significant age population (cf. section 2.3, Jones et al., 2019).

The six Kay Peak ridge samples were further evaluated by performing a kernel density estimation (Lowell, 1995). In the kernel density estimate plot (Fig. A1), the two youngest in situ $^{14}$C ages from Kay Peak (from samples KAY-105 and KAY-109), constitute a second distinctive but smaller peak. Removing these two in situ $^{14}$C ages results in a more normal distribution (Fig. A2) consistent with a single age population, implying that in situ $^{14}$C exposure ages measured from KAY-105 and KAY-109 are outliers. The removal of KAY-105 and KAY-109 provides a revised $^{14}$C based mean deglaciation age for Kay Peak ridge of 6.5 ± 0.3 ka (1SE, n=4), with a reduced $\chi^2$v = 3.29 and p value ≥ 0.01 (Table 1). This revised mean deglaciation age falls between the mean $^{10}$Be deglaciation ages for scoria cone outcrops A and B (7.1 ± 0.1 and 6.4 ± 0.1 ka, respectively). The two in situ $^{14}$C ages for KAY-105 and KAY-109 were included in the previously published linear regression analysis of Johnson et al. (2020). Further linear regression analysis was conducted to understand the impact of removing these two exposure ages on our interpretations of the thinning history of Pope Glacier. The results and implications of this sensitivity test are described in section 3.1.

## 3.1 Results of exposure age linear regression analysis

In the following section, we discuss the ice surface thinning rates and best fit constraints for end of thinning, which we define as the time that the ice surface lowered to 80 m above the modern ice surface, i.e., the elevation of the lowermost sample included in the linear regression transect. The thinning rates and best fit were calculated using the iceTEA Monte Carlo linear regression model (Jones et al., 2019). This model randomly applies a least squares regression to exposure ages which are normally distributed through a Monte Carlo simulation (Jones et al., 2019). All scoria cone samples were first evaluated for their inclusion in the revised input dataset following the principles outlined in Johnson et al., 2020, whereby samples would be removed if they 1) exhibited $^{10}$Be inheritance or 2) were anomalously young (> 2 standard deviations from the mean). Average ice surface lowering rates for Pope Glacier were then calculated using 5000 iterations of linear regression through the $^{10}$Be and $^{14}$C ages and their internal uncertainties.

We used four different age datasets for the linear regression analysis. These different input datasets permit quantification of the improvement of our new ages on the thinning history of Pope Glacier and enabled us to conduct a sensitivity analysis on the inclusion or exclusion of different age data. The four sample sets are: Sample set 1 – the original dataset used in Johnson et al., 2020, which serves as a baseline comparison for sample sets 2 – 4; sample set 2 – the Johnson et al., 2020 dataset and our 12 new $^{10}$Be exposure ages from scoria cone; sample set 3 – the Johnson et al., 2020 dataset with KAY-105 and KAY-109 $^{14}$C exposure ages removed; and sample set 4 – the Johnson et al., 2020 dataset with KAY-105 and KAY-109 $^{14}$C exposure ages removed and the 12 new $^{10}$Be exposure ages from scoria cone included.

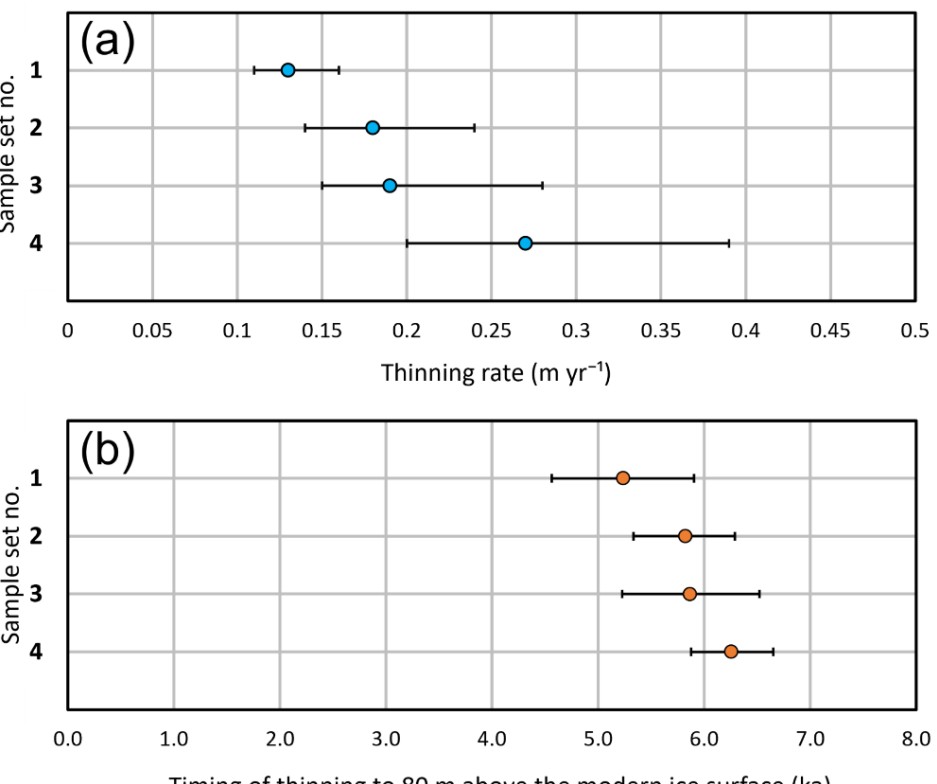

**Figure 5**: **Thinning rates and age constraints from linear regression analysis.** (a) range in thinning rates (m yr[-1]) compiled from linear regression histograms (Fig. B1) and (b) uncertainty range in best fit timing of thinning to 80 m above the modern ice surface (ka) calculated for each of the different input data to the linear regression Monte Carlo simulation (Fig. B2). Blue circles in (a) represent the median thinning rate and orange circles (b) represent the endpoint of the best fit thinning line. Error bars represent 68 % confidence interval uncertainty estimates in plot (a) and 95 % confidence interval uncertainty estimates in plot (b). The sample set no. indicates the four different age datasets

used for linear regression analysis, which are described in section 3.1. Note plot b, sample sets 1 and 4 linear regression best fit end ages correspond to the linear regression transects displayed in Figure 6.

Our preferred input dataset, sample set 4, which includes the addition of our new exposure age data from scoria cone and omission of KAY-105 and KAY-109, changes the average thinning rate from 0.13 +0.03/-0.02 m yr[-1] between ~9–5 ka (Fig.

5a) (Johnson et al., 2020) to 0.27 +0.12/-0.07 m yr[-1] between 8.1-6.3 ka (Fig. 5b). The difference in the median thinning rate is 0.14 m yr[-1], which is an increase of 52% on the previously published thinning rate. The ranges of thinning rates (Fig. 5a) derived using Sample Set 1: 0.11–0.15 m yr[-1] and Sample Set 4: 0.2–0.39 m yr[-1] do not overlap within 68 % confidence intervals. The range in thinning rates derived from Sample Set 2 of 0.14–0.24 m yr[-1], however, does overlap within the 68 % confidence interval uncertainty range with the previously published rate from Johnson et al., 2020, with an increase in the

median thinning rate of 28 %. Considered in isolation, the removal of KAY-105 and KAY-109 (Sample Set 3) increases the median rate of thinning (m yr[-1]) by 32 % compared to the data published in Johnson et al., 2020.

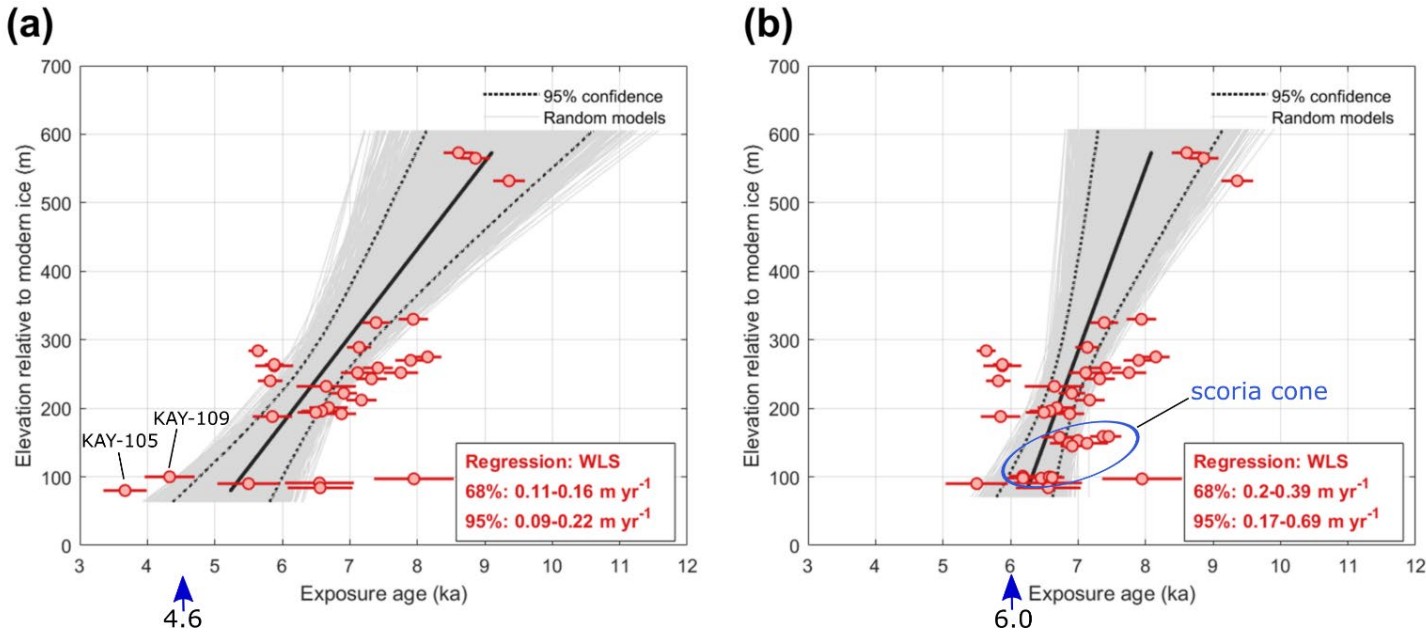

**Figure 6: Linear regression transects of ice surface lowering over time (ka) relative to the modern ice surface (m).** (a) Results of linear regression using the same exposure ages as Johnson et al. (2020). (b) Results of linear regression using new scoria cone [10]Be exposure ages
with KAY-105 and KAY-109 outliers removed. In both panels, exposure ages are plotted relative to elevation above the modern ice surface and display (1σ) external errors. Modern ice surface elevations are as reported in Johnson et al., 2020, and using 80 m asl as the modern ice surface elevation adjacent to the scoria cone site. The dotted black lines delimit the 95% confidence intervals of the thinning profile (also shown in Fig. 5b). The straight black line displays the model best fit line and grey lines represent all model fits to the data. Surface exposure age elevations were input normalised to the height above the modern ice surface at Kay Peak (80 m). The blue arrows indicate timing ice
surface reached its current elevation based on extrapolation of the best fit line for each transect. The blue ellipse in panel b indicates the position of scoria cone exposure ages on the linear regression transect. See Fig. B2 for the linear regression transects generated from Sample Sets 2 and 3.

The new [10]Be exposure ages from scoria cone better constrain the timing that Pope Glacier lowered to ~80 m above its present

elevation, where the present or modern ice surface is defined as 80 m asl adjacent to the scoria cone site. The best fit end of

thinning is changed from 5.2 ± 0.7 ka (95 % confidence interval) to 5.8 ± 0.5 ka (95 % confidence interval) using the same

input sample set as Johnson et al., 2020 plus the 12 new [10]Be ages. The omission of in situ [14]C ages from samples KAY-105

and KAY-109 further changes the best fit end of thinning and shifts the timing of ice surface lowering to ~80 m above the

modern ice elevation to 6.3 ± 0.4 ka (95 % confidence interval). The best fit onset of thinning is also shifted from 9.1 ka ± 1.1

to 8.1 ± 0.9 ka. Removing these two outliers is justified because they were shown to exhibit a non-normal distribution (Table

1, Fig. A1). When only the scoria cone ages are used in the linear regression analysis, with no re-evaluation of the Kay Peak

data, the modern ice surface is reached at 5.4 ka. Extrapolation of the Johnson et al., 2020 best fit line suggests Pope Glacier

lowered to its present elevation by 4.6 ka (Fig. 6a), whereas the revised best fit line indicates Pope Glacier lowered to its

present thickness by 6.0 ka (Fig. 6b).

Considering the complex topography at the scoria cone site (Fig. 3a), in order to investigate whether using a different, outcrop-specific measured ice surface elevation to calculate the vertical distance above the modern ice surface would impact our results, we performed a further sensitivity test. The linear regression analysis was repeated using our preferred input dataset (sample set 4) and outcrop-specific ice surface elevations measured more proximal to outcrop A and outcrop B, respectively, instead of our original representative ice surface elevation measured at a point on Pope Glacier a few hundred metres away from the scoria cone (see Appendix C, Fig. C1). Using an outcrop-specific ice surface elevation gives a best fit model timing and rate of thinning of 6.4 ka and 0.44 m yr$^{-1}$, respectively, which fall within the 95% confidence interval on our original preferred model (6.7–5.9 ka and 0.17–0.69 m yr$^{-1}$, respectively). The results of the sensitivity test confirm not only that using an outcrop-specific ice surface elevation to calculate the vertical distance above the modern ice surface does not lead to a statistically significant difference in our interpretation of the thinning history, but also that the uncertainties on our preferred model adequately capture any sensitivity to this input model parameter. Therefore, the choice of modern ice surface elevation does not significantly change our results or the implications of our preferred model.

## 4. Discussion

### 4.1 Wider context of the scoria cone [10]Be exposure ages

Here we discuss the wider context for the new thinning constraints presented above. The scoria cone site is situated 180–240 m asl, an elevation range not covered by existing surface exposure ages from Mt Murphy (Fig. 7). We interpret these new ages as reflecting the timing of deglaciation of the site. Specifically, the error weighted mean deglaciation ages of the upper outcrop A (240 m asl) of 7.1 ± 0.1 ka and the lower outcrop B (180 m asl) of 6.4 ± 0.1 ka suggest that the surface of Pope Glacier lowered by at least 60 m in less than 1000 years (see Fig. 6b), which is equivalent to a rate of 0.06 m yr$^{-1}$. The scoria cone ages are, in addition, tightly clustered with no outliers (Fig. 4a), i.e., no individual clasts appear to have been subject to prior exposure or post-depositional disturbance that would make their exposure ages more scattered or skew older or younger. This tight clustering of ages shows that the cobbles collected from each outcrop most likely experienced the same history of exposure, and hence the average age from each outcrop is thus likely to reflect its true deglaciation age (Balco, 2011). The lack of geologic scatter permits greater confidence in the error weighted mean exposure age calculated for each outcrop, and by extension, the thinning trajectory of Pope Glacier over the elevation range of 240–180 m asl. Evidence for minimal geologic scatter in the exposure ages is further strengthened by reduced $\chi^2$ values close to 1 and p values > 0.01 (see Results section, Table 1). In the context of the whole Mt Murphy age versus elevation profile (Fig. 7), the scoria cone exposure ages suggest that the ice surface over the lower nunataks thinned at a similar rate to that detected at the higher elevation sites (e.g., Icefall Nunatak), and did not slow significantly as it neared its present ice thickness.

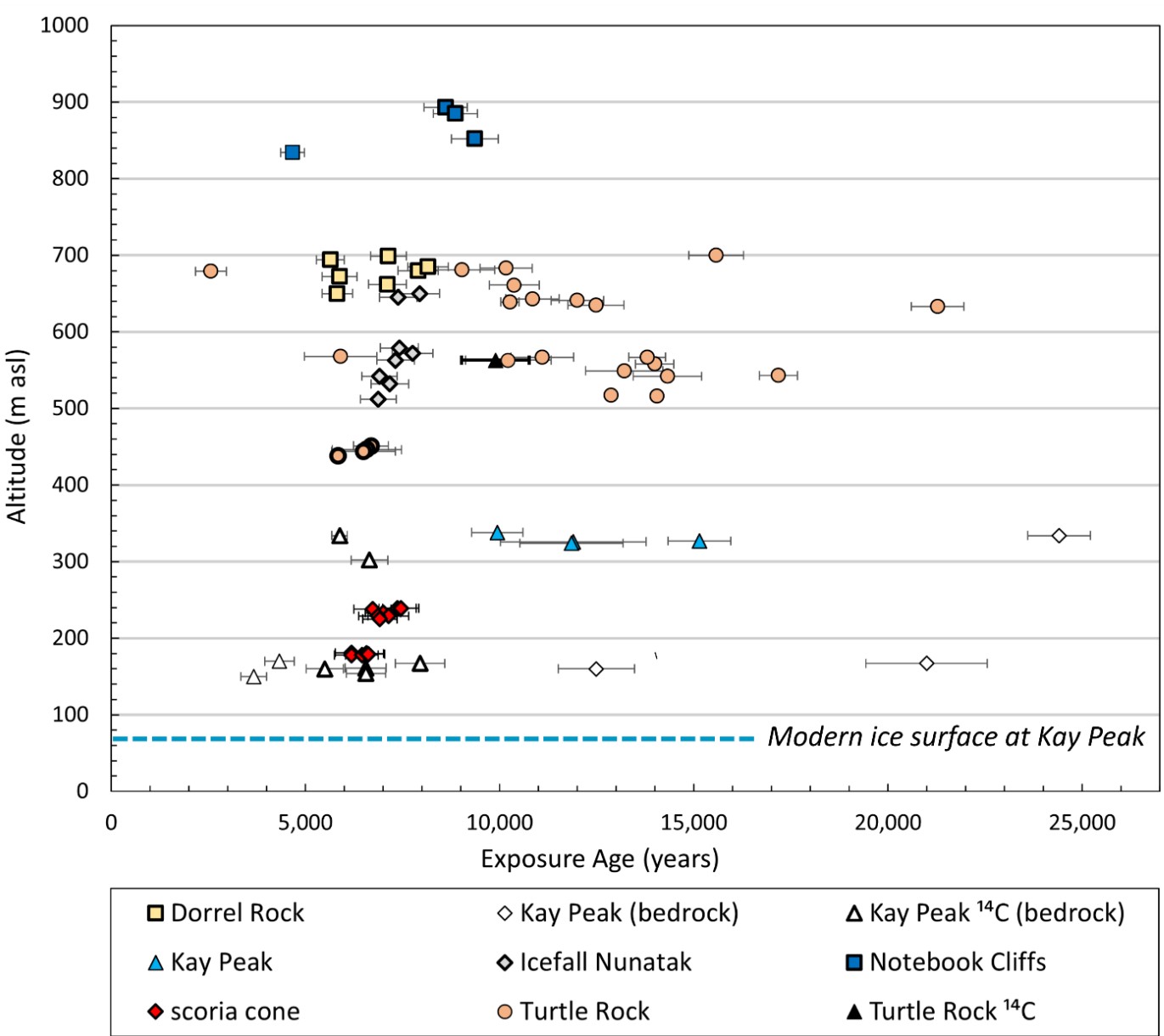

**Figure 7: New exposure ages obtained from scoria cone in comparison with exposure age data reported by Johnson et al. (2008, 2020) for nunataks surrounding Mt Murphy.** Exposure ages are plotted as [10]Be and in situ [14]C exposure ages (years) versus altitude (m asl). Filled symbols represent erratics and open symbols represent bedrock samples, all symbols represent [10]Be exposure ages unless specified as in situ [14]C in the key. Error bars represent external uncertainty in years (1σ). Bold outline indicates samples used for Sample Set 4 linear regression analysis. Dashed blue lines show the modern ice surface elevation near Kay Peak ridge (which is < 1 km from the present grounding line: Fig. 1c). Note symbols for previous and new exposure ages are the same as in Fig. 1b and Fig. 1c.

The deglacial history of the lower exposed ridges of Mt Murphy (below 300 m asl) was, until now, inferred from [10]Be and in situ [14]C cosmogenic nuclide measurements of samples from Kay Peak ridge (Fig. 1c; Johnson et al., 2020). The Kay Peak exposure ages differ from the scoria cone exposure ages in several ways. Firstly, many [10]Be Kay Peak bedrock exposure ages are considerably older than [10]Be exposure ages from erratics on the scoria cone, where the maximum deglaciation age is 7.4 ±
0.5 ka. In contrast, only one Kay Peak ridge bedrock sample analysed for [10]Be yielded an exposure age younger than the LGM (< ~20 ka). [10]Be exposure ages of Kay Peak erratics (~330 m asl) are also younger than most Kay Peak [10]Be bedrock exposure ages and all [10]Be exposure ages from erratics post-date the LGM. Measuring [10]Be concentrations in erratics is often preferred to bedrock because [10]Be inheritance is less likely in erratics due to removal of previously accumulated nuclides by glacial erosion and transport. However, the complete removal of the previous nuclide inventory is not guaranteed (Heyman et al.,
2011). Kay Peak erratic exposure ages, while younger than the LGM (15.1–9.9 ka), are still 2.5–7.5 ka older than the maximum exposure age from the scoria cone. These exposure ages would suggest Kay Peak erratics deglaciated much earlier than samples from higher elevation sites, for example Icefall Nunatak, where [10]Be exposure ages are 7.9–6.9 ka at 650–560 m asl (Johnson et al., 2020).

In addition, Kay Peak bedrock exposure ages are more scattered than the scoria cone erratic exposure ages. Some of this spread is accounted for by the greater elevation range of Kay Peak samples between 330–150 m asl. The scoria cone elevation range of 240–180 m asl is much smaller. Greater variability in Kay Peak exposure ages would therefore be expected, yet there is significantly more scatter in Kay Peak bedrock exposure ages, even over very small elevation ranges (< 20 m). Inheritance in Kay Peak [10]Be bedrock exposure ages can partly explain this scatter, but the scatter also extends to in situ [14]C exposure ages,
which limits how well we can constrain thinning closest to the modern ice surface. At lower Kay Peak ridge 170–150 m asl, individual in situ [14]C bedrock exposure ages range from 3.7 ± 0.3–8.0 ± 0.6 ka. The > 5 ka scatter in the [14]C bedrock exposure ages was speculated to be due to the complexity of snow/ice cover related to the curvature of the Kay Peak ridge crest (Johnson et al., 2020). Furthermore, there is no other location in Antarctica to date where so many samples have been measured for in situ [14]C, thus the apparent scatter in [14]C ages above that expected by analytical uncertainties alone could be in part due to
underestimation of the measurement uncertainty for [14]C concentrations.

In summary, these observations imply: i) that [10]Be ages of most of the Kay Peak ridge bedrock samples reflect inheritance of [10]Be from an earlier period of exposure, and the Kay Peak erratics were likely similarly affected (the younger in situ [14]C exposure ages from the same bedrock samples, including KAY-101 (6.0 ± 0.6 ka), KAY-107 (5.5 ± 0.6 ka), and KAY-108
(8.2 ± 0.9 ka) lend support to this hypothesis); and ii) that there is more scatter within the in situ [14]C bedrock exposure age data than in the scoria cone [10]Be erratic ages. Scatter beyond analytical uncertainty in the Kay Peak ridge [14]C exposure ages is likely primarily due to a complex localised deglaciation history caused by non-contiguous deglaciation of fringing ice along the ridge axis.

## 4.2 Implications of scoria cone exposure ages for the thinning history of Pope Glacier

In this section, we discuss the implications of our new data for the thinning history of Pope Glacier. The median thinning rate determined from the revised exposure age dataset (sample set 4; 0.27 +0.12/-0.07 m yr$^{-1}$) is the most different from the previously published median rate for Pope Glacier (0.13 +0.03/-0.02 m yr$^{-1}$; Johnson et al., 2020) of all the rates we calculated. Using this rate implies that the ice surface of Pope Glacier lowered up to 52 % faster than previously estimated. This falls in the middle of the range of thinning rates from elsewhere in Antarctica calculated using the same method (cf. Section 4, Table 3, Small et al., 2019).

Even though the mid-Holocene thinning of Pope Glacier occurred over only a few thousand years, it appears to have been much slower than contemporary changes in the region. The fastest thinning rate we calculated using the upper limit of the 95% confidence interval on our median rate (0.27 +0.12/-0.07 m yr$^{-1}$) is over an order of magnitude slower than contemporary thinning rates of 4–7 m yr$^{-1}$ detected above the 2020 grounding line of Pope Glacier (Milillo et al., 2022). However, it is important to consider the relative resolutions of the datasets. Paleo thinning rates for Pope Glacier averaged over a millennial timescale might be perceived as an oversimplification because only a single average is calculated over the period of thinning being examined. However, linear thinning rates averaged over longer timescales are thought to be more indicative of the basin average rather than localised changes in the glacier trunk (Small et al., 2019). Therefore, linear thinning rates of Pope Glacier although less sensitive to short-term fluctuations are extremely relevant for validating model simulations which are generally regional or larger in scale e.g. (Pollard et al., 2016; Johnson et al., 2021).

Our new exposure ages also have implications for the timing of the later stages of thinning, closest to the modern ice surface. The revised best fit onset and end of thinning (Fig. 8, red dotted line) from 8.1–6.3 ka indicates more abrupt thinning compared to the previously published estimate of 9.1–5.2 ka (Fig. 8, black dashed line) (Johnson et al., 2020). The trajectory of thinning indicated by the revised best fit line equates to ≥ 560 m lowering of the ice surface at Pope Glacier in approximately half the time: 1800 years compared to a previous duration of 3900 years (Johnson et al., 2020). In comparison with other parts of the ASE, this revised time span is similar to the duration of early- to mid-Holocene thinning detected at Mt Moses, in the eastern ASE (adjacent to Pine Island Glacier). The lower 142 m of presently-exposed outcrop at Mt Moses deglaciated between 7.4 ± 0.7–5.4 ± 0.7 ka, over a period of approximately 2,000 years (Johnson et al., 2014). However, at Mt Moses, the ice sheet thinned in two distinct phases at different rates. This appears to contrast with the thinning pattern observed at Mt Murphy, although the relative density of data from the two sites is not similar so fluctuations in thinning rates at Mt Moses may not have been detected. For a discussion of the paleoclimatic conditions in the ASE during the early- to mid-Holocene and their potential influence on the timing of ice surface thinning at Mt Murphy, see Johnson et al. (2020) and Sproson et al., (2022).

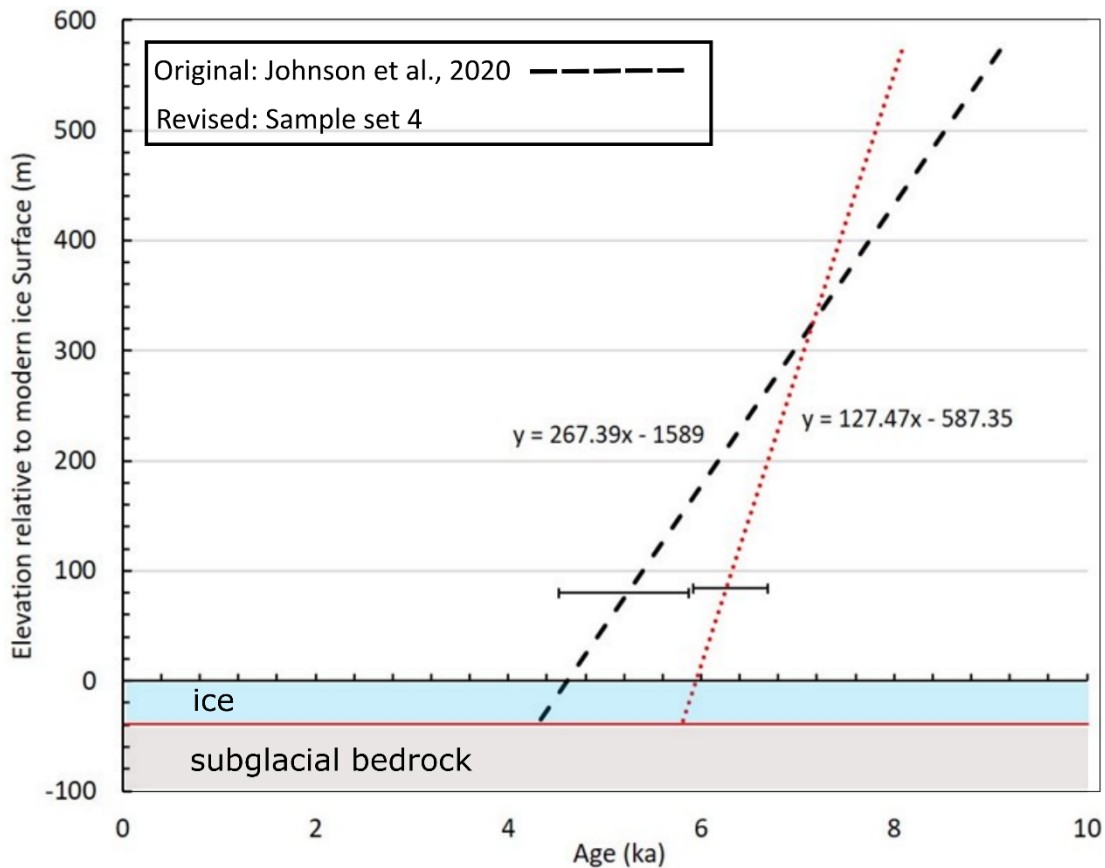

**Figure 8: Predicted timing of exposure at or near the surface of subglacial bedrock cores drilled at Kay Peak ridge.** Predictions of timing for bedrock exposure extrapolated from the two endmember best fit line thinning rate scenarios calculated from different input combinations of exposure ages. The best fit ice surface lowering rate is determined from the published Mt Murphy exposure age dataset of Johnson et al. (2020) is contrasted with the best fit from the revised dataset (sample set 4) which includes exposure ages from scoria cone and omits KAY-105 and KAY-109. Error bars represent 95 % confidence intervals of best fit timing at which the samples closest to the modern ice surface were deglaciated. Below the 95 % confidence bars the best fit line has been extrapolated based on the linear equations shown in the figure. Blue shaded area represents the ice thickness that was drilled (~40 m) to reach the subglacial bedrock. Red horizontal line is a one-dimensional representation of the bedrock surface at the point of core recovery. Note the 95 % confidence interval of best fit revised end thinning is at a slightly higher elevation (84 m) as KAY-105 is the sample at Mt Murphy closest to the modern ice surface (80 m).

Better constraints on the timing and pace of thinning of Pope Glacier during the Holocene allows us to make a prediction for the hypothetical timing of exposure of subglacial bedrock cores drilled at Kay Peak ridge (-75.215°, -110.960°). Previous work suggested that the ice surface lowered to the elevation of the modern ice (80 m asl) by 4.6 ka (Johnson et al., 2020). Performing an extrapolation on our revised best fit linear thinning history implies that Pope Glacier reached its present thickness considerably earlier in the Holocene, at 6 ka (Fig. 8). Further extrapolating our best fit ice thinning trajectory to below the

surface of the modern ice, allows us to predict when subglacial bedrock cores from below Kay Peak ridge could have been
exposed at or near to the surface. Based on the previous thinning rate estimates (Johnson et al., 2020), the subglacial bedrock from ~40 m below ice surface would have been exposed at 4.2 ka. With the new exposure ages and revised Kay Peak ages, our linear regression analysis suggests that the likely onset of exposure of the subglacial bedrock occurred ~1,500 years earlier, at 5.7 ka. However, we acknowledge the assumption of linear constant thinning is only valid over the range of elevations specified by our linear regression analysis (Fig. 5); as such, our chronology is robustly constrained only to 80 m above the
modern ice surface. Nevertheless, our results suggest that early-to mid-Holocene ice thinning at Mt Murphy occurred over a shorter interval than previously assumed and implies a longer duration over which any subsequent rethickening of ice could have occurred.

## 5. Conclusions

We present 12 new cosmogenic $^{10}$Be exposure ages which provide constraints on the timing of the last deglaciation of the
western flank of Mt Murphy, in the Amundsen Sea Embayment. The ages were derived from erratic cobbles collected from two outcrops on a scoria cone situated within ~160 m of the modern ice surface and ~1 km from the present grounding line of Pope Glacier. Outlier detection was applied to both the new exposure ages and to existing exposure ages below 300 m asl to better constrain the rate and timing of thinning of Pope Glacier during the Holocene. The new $^{10}$Be exposure ages represent two statistically distinct populations, which correspond to two rock outcrops within an elevation range of 240-180 m asl and
have an error weighted mean age and standard error of 7.1 ± 0.1 ka and 6.4 ± 0.1 ka, respectively.

Linear regression analysis undertaken for this study implies that Pope Glacier thinned by ≥ 560 m at a median rate of 0.27 m yr$^{-1}$ over a period of 1,800 years during the early- to mid-Holocene. This is 1.5 times faster than previously assumed. Furthermore, the tighter constraints placed by the new data on the timing of deglaciation of the lowest currently exposed
section of Mt Murphy suggest that the ice surface of Pope Glacier had thinned to within 80 m of its present elevation by 6.3 ± 0.4 ka. This is 1,100 years earlier than the previous estimate of Johnson et al. (2020).

These results have implications for bedrock cores collected for a parallel study from below the ice sheet near the lowermost outcrop of Kay Peak ridge, close to our study site: the revised thinning trajectory suggests that the top of those cores could
have been exposed at or near the ice sheet surface as early as 5.7 ka.  In summary, the results suggest that early- to mid-Holocene thinning of Pope Glacier occurred more rapidly, and earlier, than previously thought. They therefore permit either a longer period of ice sheet stability in the mid- to late-Holocene, or alternatively a longer duration over which late Holocene rethickening could have occurred had the ice sheet not remained stable.


## Appendices

### Appendix A

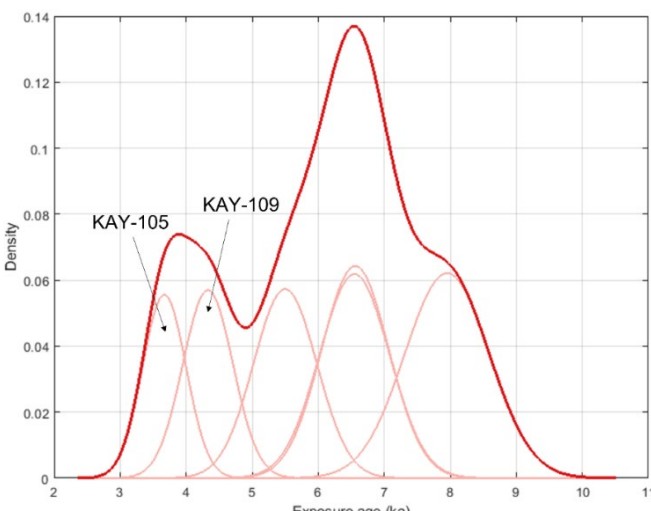

**Figure A1: Kernel density estimate plot of $^{14}$C ages (n = 6) distribution for lower Kay Peak ridge.** Error weighted mean: 5173; reduced
chi-squared: 15.37; chi-squared p-value: 0.0000

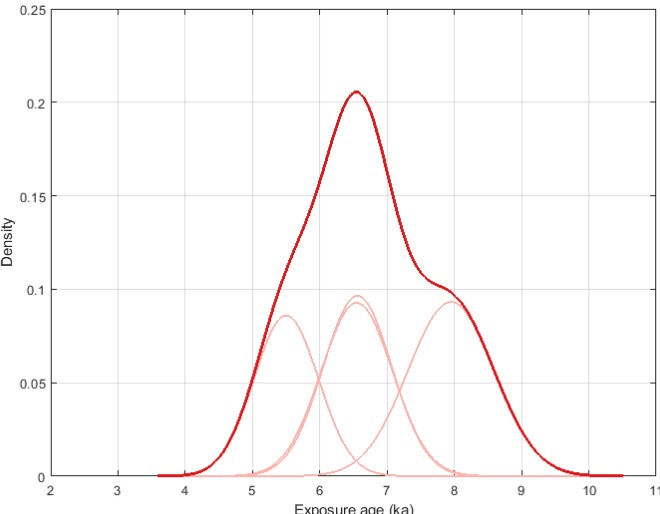

**Figure A2: Kernel density estimate plot of $^{14}$C ages distribution for lower Kay Peak Ridge (n=4), with the two youngest ages (KAY 105, KAY 109) removed**. Error weighted mean: 6472; Standard error: 266; External error: 205; reduced chi-squared: 3.29; chi-squared p-value: 0.0196

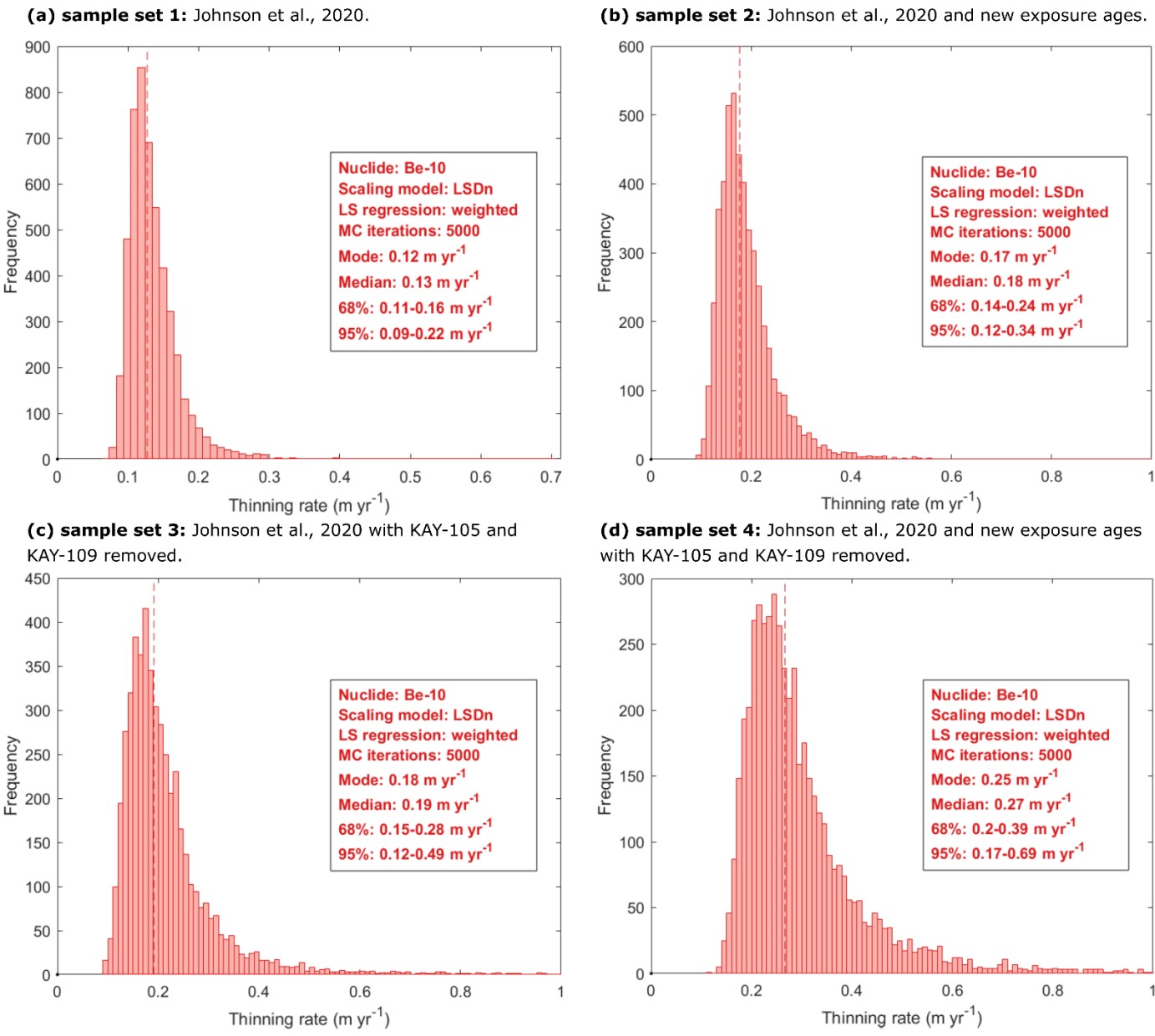

**(a) sample set 1:** Johnson et al., 2020.

**(b) sample set 2:** Johnson et al., 2020 and new exposure ages.

**(c) sample set 3:** Johnson et al., 2020 with KAY-105 and KAY-109 removed.

**(d) sample set 4:** Johnson et al., 2020 and new exposure ages with KAY-105 and KAY-109 removed.

**Figure B1: Histogram outputs of ice surface thinning rates generated by iceTEA (Jones et al., 2019).** Results of linear regression: (a) the original sample set of Johnson et al. (2020), (b) incorporating new data from Scoria Cone, (c) removing young ages identified as outliers and (d) incorporating new data from Scoria Cone as well as removing young ages identified as outliers. Panel boxes inset in each subfigure display the median and modal values for thinning rates as well as the range of 68 % and 95 % confidence intervals.

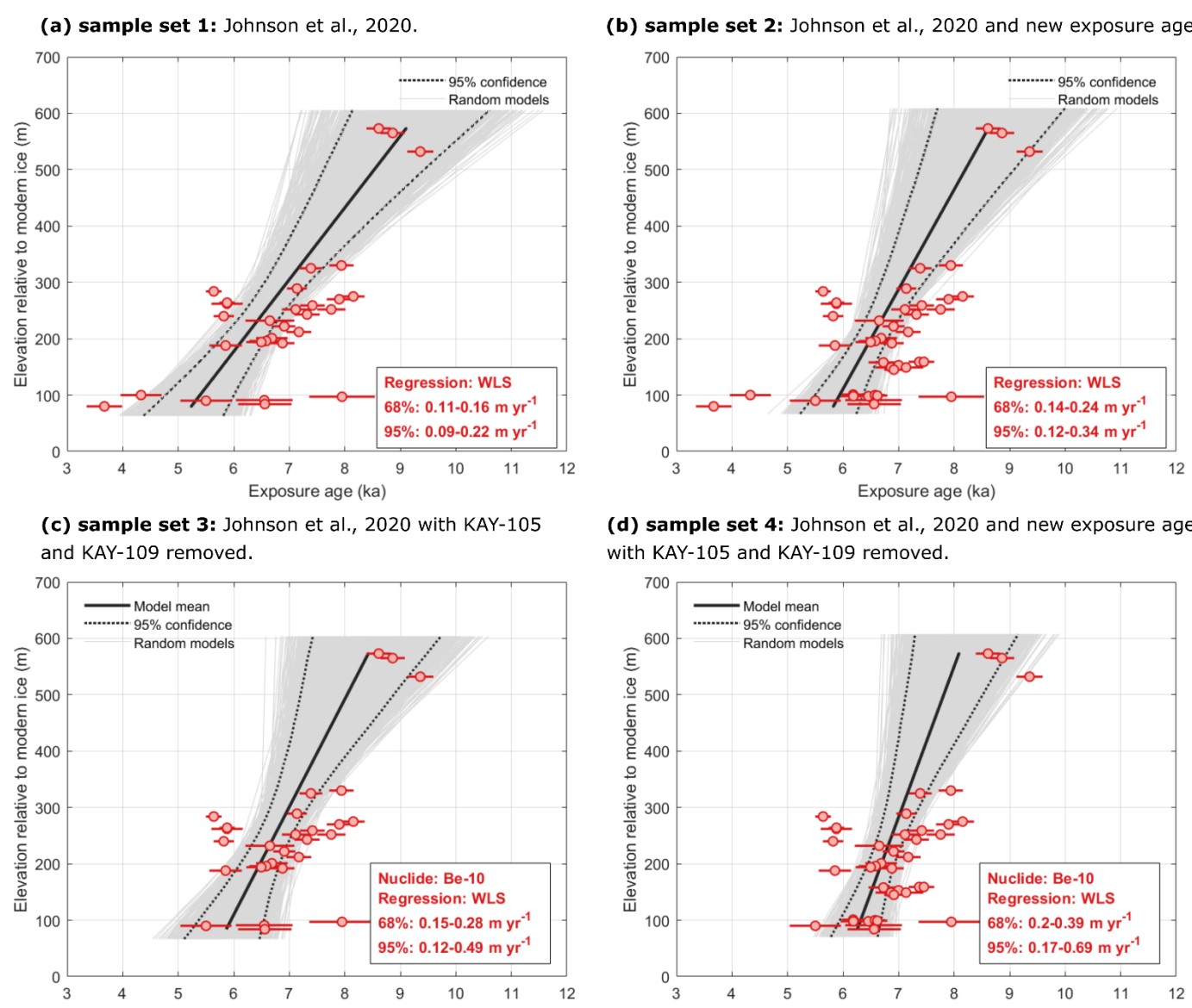

**(a) sample set 1:** Johnson et al., 2020.

**(b) sample set 2:** Johnson et al., 2020 and new exposure ages.

**(c) sample set 3:** Johnson et al., 2020 with KAY-105 and KAY-109 removed.

**(d) sample set 4:** Johnson et al., 2020 and new exposure ages with KAY-105 and KAY-109 removed.

**Figure B2: Linear Regression Transect of ice surface thinning generated with iceTEA:** Linear regression transects showing (a) the original sample set of Johnson et al. (2020), (b) incorporating new data from Scoria Cone, (c) removing young ages identified as outliers and (d) incorporating new data from Scoria Cone as well as removing young ages identified as outliers. Black solid line is 'best fit' averaged from 5000 Monte Carlo simulations. Dotted black lines represent 95 % confidence intervals. Exposure ages displayed as red circles with internal uncertainties.



**Appendix C** – Topographic profiles of scoria cone relative to modern ice surface elevation and sensitivity test results.

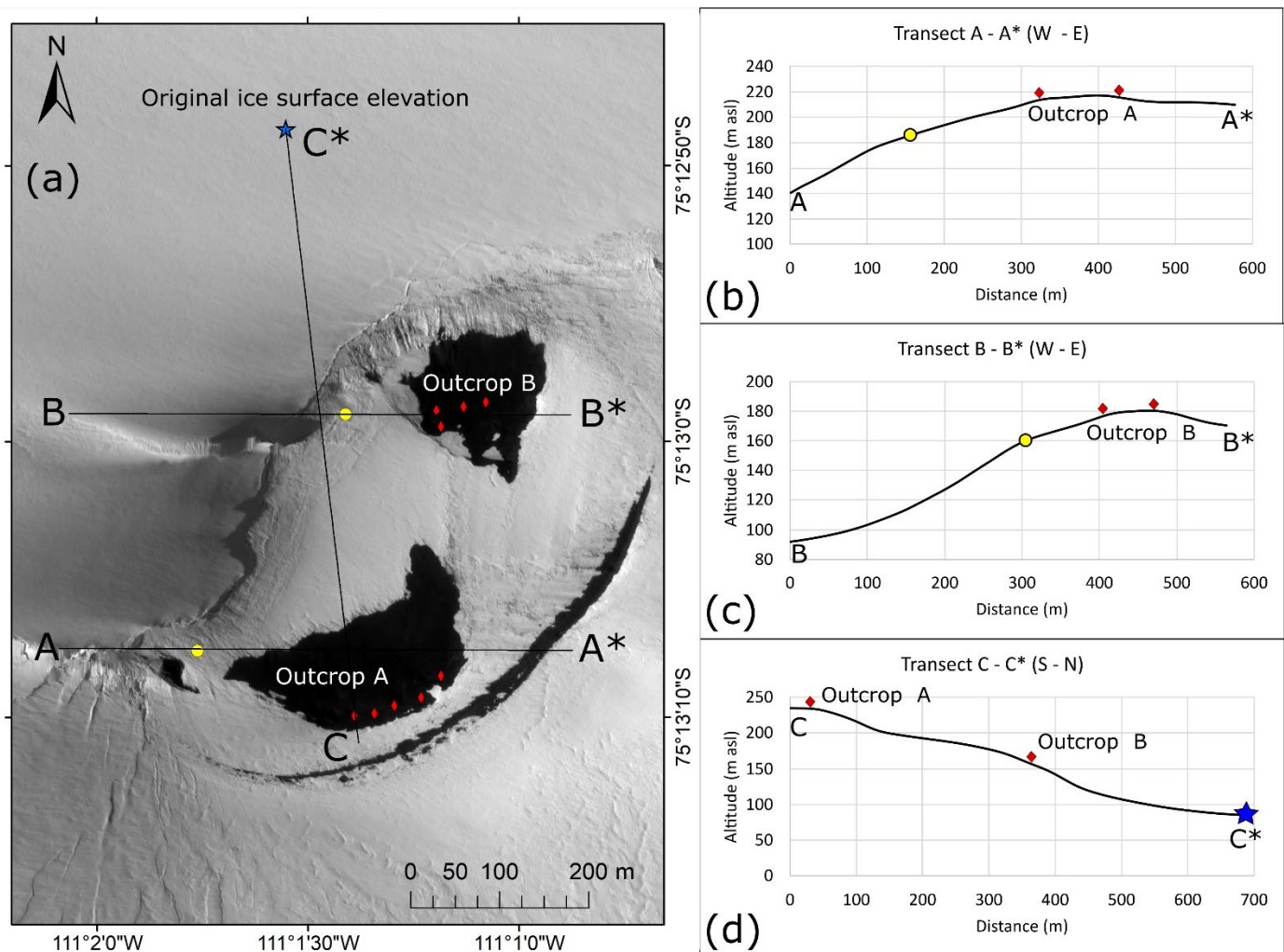

**Figure C1: Transects displaying topographic profile of scoria cone and ice surface elevations adjacent to the Pope Glacier**. **(a)** Map
showing scoria cone outcrops A and B adjacent to the Pope Glacier, sample locations, and location of topographic profiles along transects
A–A*, B–B* and C–C*. Location of original representative reference modern ice surface elevation (blue star) at 80 m asl was measured
from a Mt Murphy digital elevation model (DEM). Outcrop-specific ice surface elevations (yellow circles) used to calculate vertical distances
above the ice surface relative to outcrop A and outcrop B were input for the sensitivity test of the linear regression analyses. Red diamonds
indicate the position of scoria cone samples. **(b)** Topographic profile along transects A–A* for outcrop A with outcrop-specific ice surface
elevation (yellow circle at 183 m asl) and sample positions adjacent to transect. **(c)** Topographic profile along transects B–B* for outcrop B
with outcrop-specific ice surface elevation (yellow circle at 159 m asl) and sample positions adjacent to transect. **(d)** transect C–C* showing
topographic profile extending S–N from scoria cone outcrop A to the original representative ice surface elevation at -75.21352° / -111.02586°
that was used to calculate the vertical distance above the modern ice surface in our preferred model for ice surface thinning rate and timing
(main paper, Fig. 6b). For transect C–C*, one representative sample elevation is shown for outcrop A and outcrop B, respectively. Note
some sample locations (n = 12) are undifferentiated on the map and transects due to their close proximity.

A measured ice surface elevation of 80 m asl was originally selected as the representative modern ice surface elevation of Pope Glacier relative to scoria cone because the ice sheet surface in the vicinity of scoria cone achieves a relatively constant elevation a few hundred metres northwest of outcrop A and outcrop B (Figure C1, a, d). However, this original representative ice surface elevation value used to model our preferred thinning history (main text, Fig. 6b, Fig. 8) may not adequately reflect the exposure

history of the scoria cone samples because it does not consider the local topographic complexity of the ice surface adjacent to each outcrop. To determine if the complex local geometry of the ice surface near the scoria cone site impacts the results of our linear regression analysis for our preferred model (i.e., using sample set 4), we performed a sensitivity analysis using two outcrop-specific ice surface elevation values (Fig. C1) measured more proximally to outcrop A (183 m asl) and outcrop B (159 m asl), respectively. Using these two outcrop-specific ice surface elevations, the calculated vertical distance of samples above

the modern ice surface were ~ 40 m at outcrop A, and ~20 m at outcrop B, respectively.

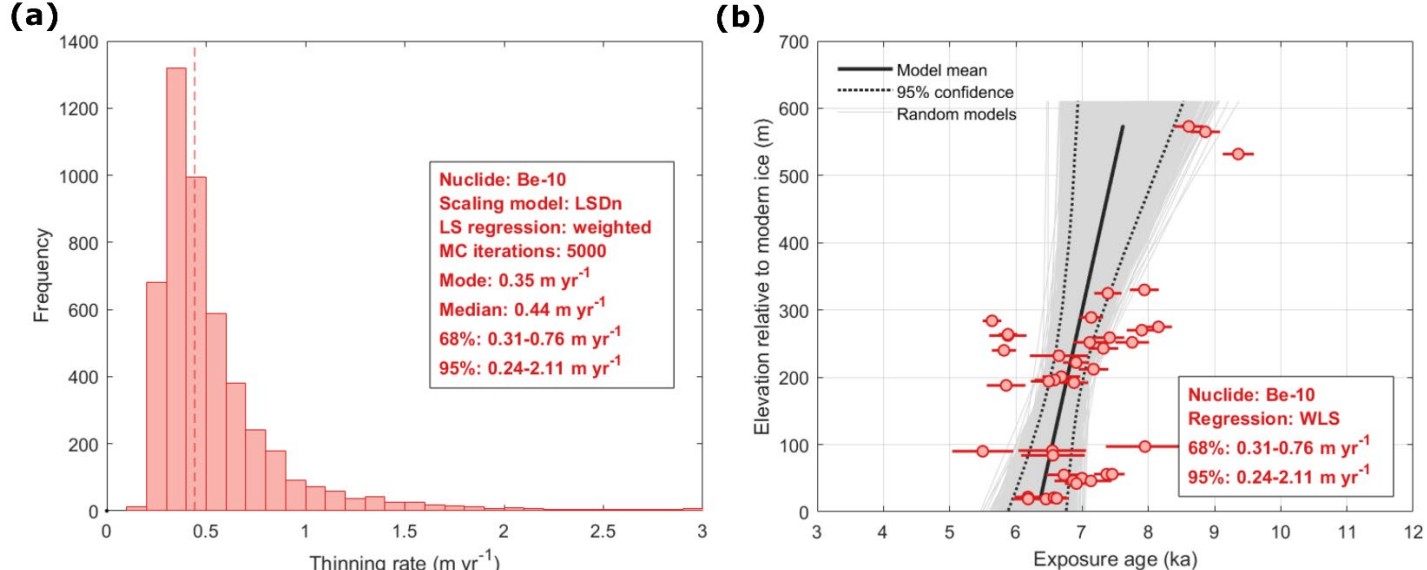

**Figure C2: Results for sensitivity test of linear regression analysis (a)** Histogram showing thinning rate output and **(b)** linear regression analysis generated by iceTEA (Jones et al., 2019) used to calculate timing and rate of ice sheet thinning. The relative elevations (vertical

distance above ice surface elevation) were calculated using outcrop-specific ice surface elevations for outcrop A and outcrop B, respectively, rather than the original measured representative ice surface elevation (80 m asl) that was used to model our preferred thinning history (main text, Figure 6, Figure 8).


| Key metric | Representative ice surface elevation (80 m asl) | Outcrop-specific ice surface elevation (outcrops A and B) |
|---|---|---|
| Median thinning rate (m yr$^{-1}$) | 0.27 | 0.44 |
| 95 % conf. int. of thinning rate (m yr$^{-1}$) | 0.17 – 0.69 | 0.24 – 2.11 |
| Best fit timing of thinning to modern ice surface (ka) | 6.3 | 6.4 |
| 95 % conf. int. of thinning to modern ice surface (ka) | 6.7 – 5.9 | 6.8 – 5.9 |


**Table C1:** Comparison of key metrics (thinning rate and timing) output from our preferred thinning history calculated from sample set 4 using a single measured representative ice surface elevation (80 m asl) to outputs from our sensitivity test calculated using outcrop-specific ice surface elevations for outcrop A and outcrop B, respectively (Figure C1).


Based on the comparison of our sensitivity test results to our original, preferred ice thinning history model (Table C1, Fig. C2), the median thinning rate calculated using outcrop-specific ice surface elevations (0.44 m yr$^{-1}$)[1] is faster than our preferred model, but falls within the 95% confidence interval of our preferred thinning rate (0.17–0.69 m yr$^{-1}$) that was derived using a measured representative ice elevation of 80 m asl. The best fit timing of deglaciation to the modern ice surface calculated using

the outcrop-specific ice surface elevations is 6.4 ka, which is slightly older than the best-fit timing for our original, preferred model (6.3 ka), i.e., the modern ice surface elevation was reached 100 years earlier based on our sensitivity test using outcrop-specific surface elevations from scoria cone. In addition, the best fit timing of deglaciation using outcrop-specific ice surface elevations (6.4 ka) also falls within the 95% confidence interval of our preferred model (6.7–5.9 ka) (main text, Fig. 5b, Fig. 6b). Therefore, based on the results of the sensitivity test, using two outcrop-specific ice surface elevations rather than a single

representative ice surface elevation does not result in a statistically significant difference in our interpretation of the ice thinning history, and we cannot reject our preferred model derived from Sample Set 4 using our original representative modern ice surface elevation of 80 m asl. Furthermore, the sensitivity test shows that our interpretation of the thinning history is insensitive, within the uncertainties of our preferred model, to our choice of ice surface elevations at scoria cone. Importantly, using the outcrop-specific ice surface elevations results in a faster median thinning rate and older timing of deglaciation, which is

consistent with our primary conclusions that early- to mid-Holocene ice surface thinning at Mt Murphy occurred at a faster rate and reached the modern ice surface earlier than previously thought.


## Data availability

Exposure age data shown in Figure 4 are publicly accessible in the UK Polar Data Centre, DOI: https://doi.org/10.5285/8F275626-5F22-48DF-95E5-CDC8F204A897


## Author Contributions

JRA led the study with supervision from JSJ, DHR, SJR, and PJM. JRA prepared the samples for AMS analysis and wrote the first manuscript draft with support from KAN, JSJ and DHR. JSJ and SJR collected the samples. JRA and RAV prepared the figures with input from all co-authors. DHR supervised analytical work and KW performed AMS analyses. All authors

contributed to writing the manuscript.

## Competing Interests

The authors declare that they have no conflict of interest.

## Acknowledgements

We appreciate the support of several people: Laura Gerrish (BAS) and Louise Ireland (BAS) for advice and feedback on satellite imagery, Mark Evans (BAS) for rock sample preparation, Richard Selwyn Jones (Monash University) for assistance with iceTEA. This work is from the "Geological History Constraints" GHC project, a component of the International Thwaites Glacier Collaboration (ITGC). Support from National Science Foundation (NSF: Grant OPP-1738989) and Natural

Environment Research Council (NERC: Grant NE/S006710/1, NE/S006753/1 and NE/K012088/1). Logistics provided by NSF-U.S. Antarctic Program and NERC-British Antarctic Survey. ITGC Contribution No. ITGC:071. We also acknowledge Scott Braddock and Seth Campbell of the GHC team for their support. Constructive reviews by Derek Fabel and an anonymous reviewer as well as helpful comments from the editor, Arjen Stroeven, greatly helped to improve the manuscript. The authors also acknowledge the financial support from the Australian Government for the Centre for Accelerator Science at the

Australian Nuclear Science and Technology Organisation (ANSTO) through the National Collaborative Research Infrastructure Strategy (NCRIS).

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
