# Peer review of "New 10Be exposure ages improve Holocene ice sheet thinning history near the grounding line of Pope Glacier, Antarctica"

_The Cryosphere, 2022_

## Author Response (AR1)

**Author's Response Document**

Replies to reviewer comments are in **BLUE** with text quoted from the revised manuscript in **RED**.

Specified line numbers in **BLACK** refer to corresponding line numbers in submitted revised manuscript.

> *We would like to thank both reviewers for their constructive and thorough reviews, and in particular Reviewer 1 for some thought-provoking comments that have helped us to improve the manuscript. Thank you also to Reviewer 2 for sharing that he enjoyed reading the manuscript – we really appreciate his encouraging comments, as well as his thoroughness detecting mistakes that had been overlooked by us, particularly regarding figure numbering. Both reviewers brought to our attention several minor issues and grammatical errors that have been corrected in the revised version of the manuscript. A point-by-point response to all reviewer comments is provided below.*

**Review #1**

Ice sheet history of Pope Glacier in Amundsen Sea embayment (ASE). Based on newly obtained Be-10 surface exposure ages and evaluation of the existing data set from Johnson et al. (2020), the authors refined the ice thinning rate and timing of deglaciation at the lowest site currently exposed. Because constraining the past ice behaviour will provide insight into the drivers and mechanisms of the rapid ice mass loss and for model validation and refinement, this research is of international scientific interest.

Although this paper makes an excellent addition to our knowledge about the Holocene ice thinning history in West Antarctic Ice Sheet, I found some points need to be clear before publication.

> *We are happy with the positive nature of this review and pleased that the Reviewer deems the manuscript a good addition to the existing knowledge of the Holocene ice thinning history of West Antarctica. The reviewer provided many helpful suggestions of how to improve and clarify our study further, which we have incorporated into the revised manuscript.*

Effect of the geometry of the ice sheet. The authors use the same value (80 m asl) as the modern ice surface elevation. However, the curvature of the ice surface around the scoria cone looks not simple and may affect the timing of the exposure of samples. Topographic profiles of the scoria cone (including outcrop A to B) and ice surface nearby should be presented. The relative height of each sample site from the contemporary ice sheet surface may be better for the thinning rate calculation.

> *The reviewer is correct that the local topography could affect the linear regression to some extent. In our original manuscript, we selected a representative measured ice surface elevation at a point on Pope Glacier only a few hundred metres away from the scoria cone where the ice stream achieves a relatively constant elevation (80 m asl). In order to address the reviewer's comment and check whether using a different, more proximal and outcrop-specific ice surface elevation to calculate the vertical distance above the modern ice surface would significantly affect the linear regression results, we performed a sensitivity test using two different outcrop-specific measured ice surface elevations: one elevation more proximal to outcrop A, and the other elevation more proximal to outcrop B. We then compared the linear regression results using the representative and outcrop-specific ice surface elevations and examined if this choice of reference ice elevation resulted in a*

*statistically significant change to the results of the linear regression analysis relative to our*
*preferred model and uncertainties. This sensitivity test showed that the linear regression*
*results using the measured outcrop-specific ice surface elevations to calculate the vertical*
*distance above the modern ice surface fell within the uncertainties of our preferred*
*thinning history using our original, preferred reference elevation of 80 m asl, and therefore*
*the choice of reference ice surface elevation did not significantly impact the main results or*
*conclusions of our study. We have added a section describing the sensitivity test and its*
*significance to Appendix C and then referenced this Appendix in the main text. We hope*
*this will help readers to better understand our choice of reference ice surface elevation and*
*the insensitivity of our conclusions to this choice of ice elevation.*

*Line 286-298: "Considering the complex topography at the scoria cone site (Fig. 3a), in*
*order to investigate whether using a different, outcrop-specific measured ice surface*
*elevation to calculate the vertical distance above the modern ice surface would impact our*
*results, we performed a further sensitivity test. The linear regression analysis was repeated*
*using our preferred input dataset (sample set 4) and outcrop-specific ice surface elevations*
*measured more proximal to outcrop A and outcrop B, respectively, instead of our original*
*representative ice surface elevation measured at a point on Pope Glacier a few hundred*
*metres away from the scoria cone (see Appendix C, Fig. C1). Using an outcrop-specific ice*
*surface elevation gives a best fit model timing and rate of thinning of 6.4 ka and 0.44 m yr$^{-1}$,*
*respectively, which fall within the 95% confidence interval on our original preferred*
*model (6.7–5.9 ka and 0.17–0.69 m yr$^{-1}$, respectively). The results of the sensitivity test*
*confirm not only that using an outcrop-specific ice surface elevation to calculate the*
*vertical distance above the modern ice surface does not lead to a statistically significant*
*difference in our interpretation of the thinning history, but also that the uncertainties on*
*our preferred model adequately capture any sensitivity to this input model parameter.*
*Therefore, the choice of modern ice surface elevation does not significantly change our*
*results or the implications of our preferred model."*

Topographic profiles of the scoria cone (including outcrop A to B) and ice surface nearby should be
presented. The relative height of each sample site from the contemporary ice sheet surface may be
better for the thinning rate calculation.

*As requested by the reviewer, in the revised manuscript, we have added a series of*
*topographic profiles, including across outcrop A and outcrop B, as well as the position of*
*scoria cone relative to our preferred reference modern ice surface elevation. These are*
*shown in Fig. C1 (Appendix C). The topographic profiles were used to inform the selection*
*of the proximal, outcrop-specific ice surface elevations used for outcrop A and outcrop B,*
*respectively, in the sensitivity test.*

[revised manuscript text omitted]

Another point to note is the measurement of sample altitudes. I do not see any description of how the
authors obtained the altitudes of the samples. If these are based on GPS measurements, the altitude
data should be corrected to Geoid highest. The difference will not be large, but it is thought to be
crucial for the interpretation with this high resolution.

*The reviewer is correct that there is no clear description of sample altitude measurement in*
*the text, and we thank them for bringing this to our attention. Sample locations were*
*recorded using a Trimble GPS 5700 receiver, that was set up as near as possible to the*
*sample and at the same height as its upper surface. Sample altitude was initially recorded*
*as height above ellipsoid and subsequently corrected to height above geoid (EGM08) in*
*metres above sea level. We have amended the text and Figure 4 caption to include a*
*description of how the sample altitudes were measured as well as further information on*
*how the reference ice height of 80 m asl was determined.*

*Line 134-140: "The samples collected from the scoria cone range in altitude from 178-239*
*m asl, which equates to an elevation of ~100-160 m above the modern ice surface. The*
*position of each sample was recorded using a Trimble 5700 GPS receiver set at the same*
*height as the sample upper surface. Height above the ellipsoid was corrected to orthometric*
*height (height above geoid EGM08) using Precise Point Positioning in Bernese software*
*(see Johnson et al., 2020). The modern ice surface elevation used in the present paper was*
*extracted from a digital elevation model (DEM) of Mt Murphy (see Johnson et al., 2020,*
*Supplementary Material). Topographic profiles illustrating the elevation and position of the*
*scoria cone outcrops and samples relative to the modern ice surface can be found in*
*Appendix C."*

*Line 178-181 (Figure caption): "The modern ice surface elevation of 80 m asl used for*
*linear thinning rate calculations was extracted from a digital elevation model (Johnson et*

*al., 2020) and has the following position: -75.21352° / -111.02586°. Note this point is ~370*
*m NW of Outcrop B and so is not visible in panel (b). For topographic profiles illustrating*
*the scoria cone outcrops relative to the modern ice surface, see Fig. C1 (Appendix)."*

Origin of the faster ice thinning. I think the refined ice sheet history probably requires some revisions
for the interpretation done by Johnson et al. (2020). Could you address this by adding a discussion
about the paleoclimatic context for Holocene thinning in ASE?

*The cause of the rapid Holocene ice sheet thinning in this region is presently unknown,*
*although some possibilities were discussed in association with the wider Holocene*
*paleoclimatic context of the region by Johnson et al. (2020), and more recently by Sproson*
*et al., (2022). Since both papers provide a detailed discussion of the topic of possible drivers*
*of early- to mid-Holocene deglaciation in the ASE, we have chosen not to repeat that work*
*here, but to instead include specific reference to both Johnson et al. (2020) and Sproson et*
*al. (2022), in our revised discussion section.*

*Line 389-390: "For a discussion of the paleoclimatic conditions in the ASE during the*
*early- to mid-Holocene and their potential influence on the timing of ice surface thinning*
*at Mt Murphy, see Johnson et al. (2020) and Sproson et al. (2022)."*

***Reviewer 1 – minor issues***

The geological background of the scoria cone should be mentioned. What are their age and origin?
And also, "bedrock surface at a scoria cone" (in the caption of Fig.3) sounds a little bit awkward for
me.

*The eruptive age of the scoria cone outcrops is not known because the bedrock has not*
*been dated. We have added a short sentence to the manuscript to clarify this. A brief*
*geological description of the outcrops was already included, so we have added a short*
*statement to clarify that they form a parasitic cone. We also amended the phrasing of*
*"bedrock surface at scoria cone".*
*Line 126-127: "The outcrops form a basaltic landform of unknown age that is a parasitic*
*cone on the main Mt Murphy volcanic shield."*
*Line 129-130 (Figure Caption): "Figure 3: Geomorphic difference between clasts deposited*
*at the scoria cone site. (a) Image showing location of scoria cone site in relation to Kay*
*Peak ridge."*

Description about the arcuate ridge landform is preferable. What is the origin of this? It looks like a
moraine ridge might be formed by readvance. Could you discuss the origin of this?

*We appreciate the reviewer's interest in the arcuate landform, and we are currently*
*investigating its age, origin, and potential record of readvance for a future paper; however,*
*a detailed discussion of this deposit is not directly relevant to our primary conclusions in*

*the present manuscript, and we thus feel it is beyond the scope of this paper. Therefore, we*
*have decided not to add any additional information about this landform in order to not*
*distract from the main focus of the paper, which is improving the mid-Holocene ice surface*
*thinning history of the lowest elevation sites at Mt Murphy.*

Line 315: Delete pace between "7." and "5"

***Done***

***Line 335: "are still 2.5 - 7.5 ka older than the maximum exposure age from the scoria***
***cone."***

Figure 6: Please make clear the origin of samples (which ones are from the Scoria cone?)

*The positions of sample exposure ages from scoria cone have been circled with a blue*
*ellipse on Fig 6b. and do not feature on Fig. 6a. The Figure caption for Figure 6 has been*
*amended to reflect this.*

[Figure]

***Line 271-272 (Figure caption): "The blue ellipse in panel b indicates the position of scoria***
***cone exposure ages on the linear regression transect."***

Table 1: uniform the number of digits for the site coordinates. I think the number of digits exceeds the
precision of the measurement (Needs more info about this).

*We have made the digits uniform to five decimal places in the revised versions of*
*Supplementary Table 1 and Supplementary Table 2. The site coordinates in decimal*
*degrees are now well within the precision of the latitude, longitude position measurements*
*obtained using the Trimble GPS.*

*Supplementary Table 1, with Latitude and Longitude (DD) coordinates displayed to within 5 decimal places.*

**Table S1**

$^{10}$Be analytical data for calculating exposure ages

| Sample ID | BAS ID | AMS ID | Latitude | Longitude | Altitude | Sample Thickness | Quartz weight | $^{9}$Be carrier | AMS measured ratio $^{10}/^{9}$Be atoms | AMS measured 1σ uncertainty $^{10}/^{9}$Be atoms | $^{10}$Be conc. | 1σ error | Blank used | $^{10}$Be/$^{9}$Be standard |
|---|---|---|---|---|---|---|---|---|---|---|---|---|---|---|
| | | (Cathode) | DD | DD | (m a.s.l.) | (cm) | (g) | (g) | | | (at.g$^{-1}$) | (at.g$^{-1}$) | | |
| CIN-101 | R15.8.1 | XBE0971 | -75.21943 | -111.02317 | 239 | 4.29 | 35.228 | 0.00025548 | 1.06E-13 | 2.62E-15 | 49987 | 1284 | BLK140920A | 07KNSTD |
| CIN-102 | R15.8.2 | XBE0972 | -75.21943 | -111.02316 | 239 | 3.09 | 35.066 | 0.00025601 | 1.07E-13 | 2.58E-15 | 51054 | 1273 | BLK140920A | 07KNSTD |
| CIN-103 | R15.8.3 | XBE0973 | -75.21941 | -111.02237 | 238 | 2.88 | 18.634 | 0.00025578 | 5.26E-14 | 1.96E-15 | 45924 | 1827 | BLK140920A | 07KNSTD |
| CIN-104 | R15.8.4 | XBE0974 | -75.21933 | -111.02158 | 233 | 3.77 | 35.049 | 0.00025646 | 9.87E-14 | 2.48E-15 | 47026 | 1228 | BLK140920A | 07KNSTD |
| CIN-105* | R15.8.5 | XBE0975 | -75.21925 | -111.02053 | 229 | 3.11 | 10.136 | 0.00025532 | 3.12E-14 | 1.19E-15 | 48248 | 2093 | BLK140920A | 07KNSTD |
| CIN-106 | R15.8.6 | XBE0976 | -75.21925 | -111.02053 | 229 | 4.16 | 13.431 | 0.00025684 | 3.85E-14 | 1.41E-15 | 45964 | 1865 | BLK140920A | 07KNSTD |
| CIN-107 | R15.8.7 | XBE0978 | -75.21903 | -111.01974 | 225 | 3.64 | 35.064 | 0.00025593 | 9.80E-14 | 2.36E-15 | 46319 | 1170 | BLK140920B | 07KNSTD |
| CIN-108 | R15.8.8 | XBE0979 | -75.21652 | -111.01973 | 181 | 3.33 | 27.756 | 0.00025654 | 6.73E-14 | 2.05E-15 | 39687 | 1291 | BLK140920B | 07KNSTD |
| CIN-109* | R15.8.9 | XBE0980 | -75.21636 | -111.01992 | 178 | 5.65 | 17.952 | 0.00025661 | 4.37E-14 | 1.46E-15 | 38851 | 1450 | BLK140920B | 07KNSTD |
| CIN-110 | R15.8.10 | XBE0981 | -75.21636 | -111.01992 | 178 | 5.11 | 32.685 | 0.00025654 | 8.07E-14 | 2.09E-15 | 40742 | 1119 | BLK140920B | 07KNSTD |
| CIN-111 | R15.8.11 | XBE0982 | -75.21632 | -111.01885 | 180 | 3.31 | 35.226 | 0.00025631 | 8.98E-14 | 2.43E-15 | 42192 | 1200 | BLK140920B | 07KNSTD |
| CIN-112 | R15.8.12 | XBE0983 | -75.21628 | -111.01796 | 179 | 2.75 | 35.130 | 0.00025631 | 9.05E-14 | 2.36E-15 | 42628 | 1166 | BLK140920B | 07KNSTD |

**Process Blanks**

| Blank | Blank ID | AMS ID | Quartz weight | 9Be carrier | AMS measured ratio $^{10}/^{9}$Be atoms | AMS 1σ uncertainty $^{10}/^{9}$Be atoms | 10Be | 1σ error | 10Be/9Be standard |
|---|---|---|---|---|---|---|---|---|---|
| | | Cathode | (g) | (g) | | | (atoms) | (atoms) | |
| A | BLK140920A | XBE0970 | 0 | 0.00025365 | 2.51E-15 | 3.75E-16 | 1.68E+08 | 6.35E+03 | 07KNSTD |
| B | BLK140920B | XBE0977 | 0 | 0.00025616 | 3.04E-15 | 4.14E-16 | 2.03E+08 | 7.09E+03 | 07KNSTD |

Scoria Cone (CIN) Be samples and process blanks BLK140920A/B were prepared for analysis at the CosmIC labs, Imperial College London. AMS analysis was performed at ANSTO, Australia.

Be-10/Be-9 measurements are "normalized to the KN-5-3 standard with an assumed ratio of 6.320 x 10^-12 ( (t1/2=1.36 Ma, Nishiizumi et al., 2007)".

 * - These samples were reprocessed because the originals were discarded due to suspected contamination.

*Supplementary Table 2, with Latitude and Longitude (DD) coordinates displayed to within 5 decimal places.*

**Table S2**
**Geomorphic data**

| Sample ID | BAS ID | Latitude DD | Longitude DD | Altitude (m a.s.l.) | Type | Lithology | Shielding Factor | Shape description | Shape elongate (1) - spherical (5) | Shape prolate (1) - equant (3) | Dimensions Long axis (cm) | Medium axis (cm) | Short axis (cm) | Weathering Classification |
|---|---|---|---|---|---|---|---|---|---|---|---|---|---|---|
| CIN-101 | R15.8.1 | -75.21943 | -111.02317 | 239 | erratic | gneiss | 0.9998 | subangular | 1 | 1 | 15 | 12 | 9 | 3 |
| CIN-102 | R15.8.2 | -75.21943 | -111.02316 | 239 | erratic | gneiss | 0.9989 | subangular - subrounded | 1 | 1 | 30 | 14 | 13 | 2 |
| CIN-103 | R15.8.3 | -75.21941 | -111.02237 | 238 | erratic | granite | 0.9998 | sub angular | 1 | 1 | 13 | 8 | 6 | 3 |
| CIN-104 | R15.8.4 | -75.21933 | -111.02158 | 233 | erratic | aplite | 0.9954 | sub rounded | 4 | 3 | 10 | 9 | 9 | 3 |
| CIN-105 | R15.8.5 | -75.21925 | -111.02053 | 229 | erratic | granite | 0.9996 | angular - sub angular | 2 | 1 | 13 | 7.5 | 6 | 1 - 2 |
| CIN-106 | R15.8.6 | -75.21925 | -111.02053 | 229 | erratic | gneiss | 0.9997 | subrounded | 2 | 2 | 22 | 16 | 8 | 3 |
| CIN-107 | R15.8.7 | -75.21903 | -111.01974 | 225 | erratic | gneiss | 0.9994 | subrounded | 4 | 2 | 13 | 11.5 | 5 | 1 -2 |
| CIN-108 | R15.8.8 | -75.21652 | -111.01973 | 181 | erratic | granite | 0.9994 | subrounded | 2 | 1 | 10 | 8 | 6 | 2 |
| CIN-109 | R15.8.9 | -75.21636 | -111.01992 | 178 | erratic | gneiss | 0.9995 | subrounded | 3 | 2 | 25 | 22 | 11 | 3 |
| CIN-110 | R15.8.10 | -75.21636 | -111.01992 | 178 | erratic | aplite | 0.9995 | subangular | 2 | 2 | 19 | 14 | 8 | 2 - 3 |
| CIN-111 | R15.8.11 | -75.21632 | -111.01885 | 180 | erratic | gneiss | 0.9992 | angular | 2 | 1 | 13 | 8 | 7 | 2 |
| CIN-112 | R15.8.12 | -75.21628 | -111.01796 | 179 | erratic | aplite | 0.9994 | subangular | 3 | 2 | 20 | 18 | 15 | 2 - 3 |

NB = weathering classification

= Heavily weathered, surrounded by spallation products; no iron staning or pitting on the upper surface.
= Moderately weathered surfaces, iron stained, but flaky in parts with some spalling/ pitting of the upper surface.
3 = Intact slightly weathered or unweathered, unspalled, some with well developed weathering rind / dark up to 1 - 3 cm on exposed surfaces.

**Review #2**

The authors present 12 new 10Be surface exposure ages from glacial erratics collected on scoria cones at the northern extent of Mt Murphy, close to the grounding line of Pope Glacier which drains into Crosson Ice Shelf in the Amundsen Sea Embayment. The new ages allow the authors to improve the previously published Holocene ice sheet lowering rates from cosmogenic nuclide data obtained from the area, concluding that lowering was more rapid by a factor of about 1.5 and occurred about 1100 years earlier than previously established.

Overall this is a very good paper. It is clearly written. It presents new data that fill a gap in the existing vertical profile data at Mt Murphy. The figures are clear and necessary, although the figure numbering does not match the numbers in the text and Supplementary Material.

> *We thank Dr. Derek Fabel for his considerate and encouraging comments and are pleased to hear he enjoyed the read. Also, we appreciate his thoroughness reading through the paper and detecting mistakes that had been overlooked, particularly relating to figure numbering.*

*Reviewer 2 - Minor issues*

In Figure 5 caption at line 242, (Fig. A3) should be (Fig. B1), and at line 243 (Fig. A4) should be (Fig. B2).

> *We thank the reviewer for pointing this out. We have changed the figure numbers accordingly.*

> *Line 246-248 (Figure caption): "Figure 5: Thinning rates and age constraints from linear regression analysis. (a) range in thinning rates (m yr$^{-1}$) compiled from linear regression histograms (Fig. B1) and (b) uncertainty range in best fit timing of thinning to 80 m above the modern ice surface (ka) calculated for each of the different input data to the linear regression Monte Carlo simulation (Fig. B2)."*

There is a full stop missing in line 251.

> *Done*

> *Line 256: "to 0.27 +0.12/-0.07 m yr$^{-1}$ between 8.1 - 6.3 ka (Fig. 5b)."*

In the text at line 288 Fig. 5a should be Fig. 4a.

> *Done*

> *Line 308: "in addition, tightly clustered with no outliers (Fig. 4a)"*

At line 380 the word "is" after Mt murphy should be deleted.

> *Done*

> *Line 401 (Figure caption): "KAY-105 is the sample at Mt Murphy closest to the modern ice surface"*

At line 471 the ITGC Contribution number should be added.

*Done*

*Line 550: "NSF-U.S. Antarctic Program and NERC-British Antarctic Survey. ITGC*
*Contribution No. ITGC:071."*

***Authors note - Additional changes not requested by Reviewers***

*All instances of "in-situ" have been changed to "in situ"*

*"Early- to mid-Holocene" is used consistently throughout.*

*Numeric ranges i.e. "240–180 m asl" have been changed from hyphen to en dash.*

*The affiliation and contact email of a co-author have been updated in the authors list.*

*Line 8: "3 Department of Geology and Geological Engineering, Colorado School of Mines,*
*Golden CO 80401, USA"*

*Line 25: "Ryan A. Venturelli – venturelli@mines.edu"*

*Outcrop altered to lowercase "outcrop" to make text consistent throughout.*

*Line 186: "ages from each outcrop (outcrop A: $\chi^2 v$ = 1.67, p value $\geq$ 0.01; outcrop B: $\chi^2 v$"*

*Line 189: "these statistical analyses are consistent with the interpretation that the ages*
*from outcrop A and B are two statistically"*

*A DOI reference number has now been provided under Data Availability.*

*Line 532: "Exposure age data shown in Figure 4 are publicly accessible in the UK Polar*
*Data Centre, DOI: https://doi.org/10.5285/8F275626-5F22-48DF-95E5-CDC8F204A897"*

*Full stop added in acknowledgements section.*

*Line 551: "We also acknowledge Scott Braddock and Seth Campbell of the GHC team for*
*their support."*

*Johnson et al., 2021b reference updated to Johnson et., 2022*

*Line 630-632: Johnson, J. S., Venturelli, R. A., Balco, G., Allen, C. S., Braddock, S.,*
*Campbell, S., Goehring, B. M., Hall, B. L., Neff, P. D., Nichols, K. A., Rood, D. H.,*
*Thomas, E. R., and Woodward, J.: Review article: Existing and potential evidence for*
*Holocene grounding line retreat and readvance in Antarctica, 16, 1543–1562,*
*https://doi.org/10.5194/TC-16-1543-2022, 2022.*

*An additional reference has been provided to address paleoclimatic context.*

*Line 702-703: "Sproson, A. D., Yokoyama, Y., Miyairi, Y., Aze, T., and Totten, R. L.:*
*Holocene melting of the West Antarctic Ice Sheet driven by tropical Pacific warming, Nat.*
*Commun., 13, https://doi.org/10.1038/s41467-022-30076-2, 2022."*

*Appendix B1 histogram and Appendix B2 linear transect composite figures remade to fix*
*lines displaying in manuscript PDF.*

*Line 448-462:*